# Cellular taxonomy and spatial organization of the murine ventral posterior hypothalamus

Laura E Mickelsen[1,2†‡], William F Flynn[3†], Kristen Springer[1], Lydia Wilson[1], Eric J Beltrami[1], Mohan Bolisetty[3§], Paul Robson[3,4,5*], Alexander C Jackson[1,2,5*]

[1]Department of Physiology and Neurobiology, University of Connecticut, Storrs, United States; [2]Connecticut Institute for the Brain and Cognitive Sciences, Storrs, United States; [3]The Jackson Laboratory for Genomic Medicine, Farmington, United States; [4]Department of Genetics and Genome Sciences, University of Connecticut Health Center, Farmington, United States; [5]Institute for Systems Genomics, University of Connecticut, Farmington, United States

*For correspondence:
paul.robson@jax.org (PR);
alexander.jackson@uconn.edu
(ACJ)

[†]These authors contributed
equally to this work

Present address: [‡]National
Institute of Diabetes and
Digestive and Kidney Diseases
(NIDDK), Bethesda, United
States; [§]Bristol-Myers Squibb,
Pennington, United States

Competing interest: See
page 28

Reviewing editor: Joel K
Elmquist, University of Texas
Southwestern Medical Center,
United States

**Abstract** The ventral posterior hypothalamus (VPH) is an anatomically complex brain region implicated in arousal, reproduction, energy balance, and memory processing. However, neuronal cell type diversity within the VPH is poorly understood, an impediment to deconstructing the roles of distinct VPH circuits in physiology and behavior. To address this question, we employed a droplet-based single-cell RNA sequencing (scRNA-seq) approach to systematically classify molecularly distinct cell populations in the mouse VPH. Analysis of >16,000 single cells revealed 20 neuronal and 18 non-neuronal cell populations, defined by suites of discriminatory markers. We validated differentially expressed genes in selected neuronal populations through fluorescence in situ hybridization (FISH). Focusing on the mammillary bodies (MB), we discovered transcriptionally-distinct clusters that exhibit neuroanatomical parcellation within MB subdivisions and topographic projections to the thalamus. This single-cell transcriptomic atlas of VPH cell types provides a resource for interrogating the circuit-level mechanisms underlying the diverse functions of VPH circuits.

## Introduction

The ventral posterior hypothalamus (VPH) is a functionally and cytoarchitecturally complex region of the hypothalamus, dominated by the mammillary bodies (MB), a discrete diencephalic structure on the basal surface of the VPH. Surrounding VPH subregions include the premammillary (PM), supra-mammillary (SUM), and tuberomammillary (TMN) nuclei as well as the caudal arcuate nucleus (Arc) and caudal lateral hypothalamic area (LHA). These subregions are embedded within diverse neural systems, with widespread afferents and efferents, known to regulate distinct physiological and behavioral functions (*Saper and Lowell, 2014*; *Simerly, 2015*; *Card and Swanson, 2013*). The MB are best known as the diencephalic branch of the classic limbic Circuit of Papez (*Papez, 1937*), which links the hippocampal formation with the anterior thalamus and midbrain/pons, and is critical for spatial memory in both rodent and primate models, as well as episodic memory in humans (*Vann and Aggleton, 2004*; *Vann, 2010*; *Aggleton et al., 2010*; *Dillingham et al., 2015*; *Vann and Nelson, 2015*). VPH subregions surrounding the MB are also robust modulators of behavioral state. The ventral and dorsal PM nuclei are implicated in reproductive and defensive behaviors, respectively (*Canteras et al., 2008*; *Donato and Elias, 2011*; *Leshan and Pfaff, 2014*). The SUM is associated with arousal and modulation of theta rhythms (*Luppi et al., 2017*; *Pan and McNaughton, 2004*; *Vertes, 2015*). The Arc is a crucial node in the regulation of hunger, satiety, and reproduction

(*Graebner et al., 2015*; *Andermann and Lowell, 2017*; *Lehman et al., 2010*). Finally, the TMN, populated by histamine (HA)-synthesizing neurons, is an important modulator of wakefulness (*Brown et al., 2001*; *Haas et al., 2008*; *Panula and Nuutinen, 2013*).

The functional diversity of the VPH is likely explained by cellular heterogeneity among these neuronal populations, the neural circuits they give rise to, and the complex brain-wide networks in which they are embedded. A significant obstacle in understanding the circuit-level mechanisms underlying its function is that VPH neuronal cell type diversity is poorly understood. Here we employ single-cell RNA sequencing (scRNA-seq) to develop a comprehensive molecular census of transcriptionally distinct cell populations in the VPH of both male and female juvenile mice. In our unsupervised analysis of over 16,000 isolated single cells, we identify 18 non-neuronal clusters and 20 distinct neuronal clusters, the majority of which are glutamatergic. Transcriptionally distinct neuronal populations are defined by the unique expression of discriminatory markers that include neuropeptides, transcription factors, calcium-binding proteins, and signaling molecules. We went on to validate differentially expressed genes in a selection of identified neuron populations through multiplexed fluorescence in situ hybridization (FISH), ISH data from the Allen Brain Institute (ABA) (*Lein et al., 2007*), as well as anterograde tract-tracing in genetically-distinct populations in the MB. Taken together, our identification of the population structure and cellular diversity of VPH cell populations provides a resource for detailed genetic dissection of VPH circuits and interrogation of their specific roles in behavior, in both health and disease states.

## Results

### Isolation of single cells from the mouse VPH for transcriptomic analysis

To isolate single cells from the mouse VPH for scRNA-seq analysis, we microdissected the region of the VPH from fresh brain slices obtained from a total of five male and five female C57BL/6 mice (P30-34), as described previously (*Mickelsen et al., 2017*; *Mickelsen et al., 2019*), in two separate harvests (see Materials and methods). Single-cell suspensions were loaded onto a Chromium Controller (*Figure 1a*) and processed using the 10x Genomics platform (*Zheng et al., 2017*). All VPH microdissections were mapped for consistency using anatomical landmarks across the rostrocaudal axis (*Figure 1b*; *Paxinos, 2012*). The two harvests used 10x V2 and V3 chemistry, respectively, and both consisted of separate male and female pools for a total of four separate pools. We found little sex-dependent differences within each harvest (*Figure 1c*) and samples from the two harvests were pooled and batch corrected to account for chemistry-dependent differences (*Figure 1—figure supplement 1a*; see Materials and methods). In our pooled data set, the median transcripts (UMIs)/cell was 8,336 and the median genes/cell was 3,415 (*Figure 1d*). We used the 1,500 genes with the highest normalized dispersion for dimensionality reduction using principal component analysis (PCA) and uniform manifold approximation and projection (UMAP) followed by cluster identification using Leiden community detection, identifying a total of 20 clusters in the first iteration of clustering (*Figure 1e,f*). Neuronal and non-neuronal clusters were segregated using a two-component Gaussian mixture model trained on the per-cluster average expression of four pan-neuronal markers *Snap25*, *Syp*, *Tubb3*, and *Elavl2* (*Figure 1—figure supplement 1b,c*) leading to a binary classification of neuronal and non-neuronal cells (*Figure 1e,f*). Subsequent clustering of only neuronal cells (20 clusters; *Figure 1—figure supplement 2a,c*) and only non-neuronal cells (18 clusters; *Figure 1—figure supplement 2b,d*) showed comparable proportions from each sex and batch.

### Identification of marker genes and testing gene specificity with a classification model

Throughout this study, we present multiple links between the transcriptional signatures of groups of single cells and their anatomical mapping. A key way in which we represent the shared transcriptional signature of groups of cells is through the presentation of marker genes, the expression of which is over-represented in a given population, as violin plots, which we use to show the distribution of transcripts per cell in each cluster for a given gene. While an individual marker gene may not be sufficient to uniquely describe an individual population, we provide an exhaustive set of markers that, when used in aggregate, specifically and uniquely identify each population described herein. Given that the median number of transcripts per cell is a small multiple of the

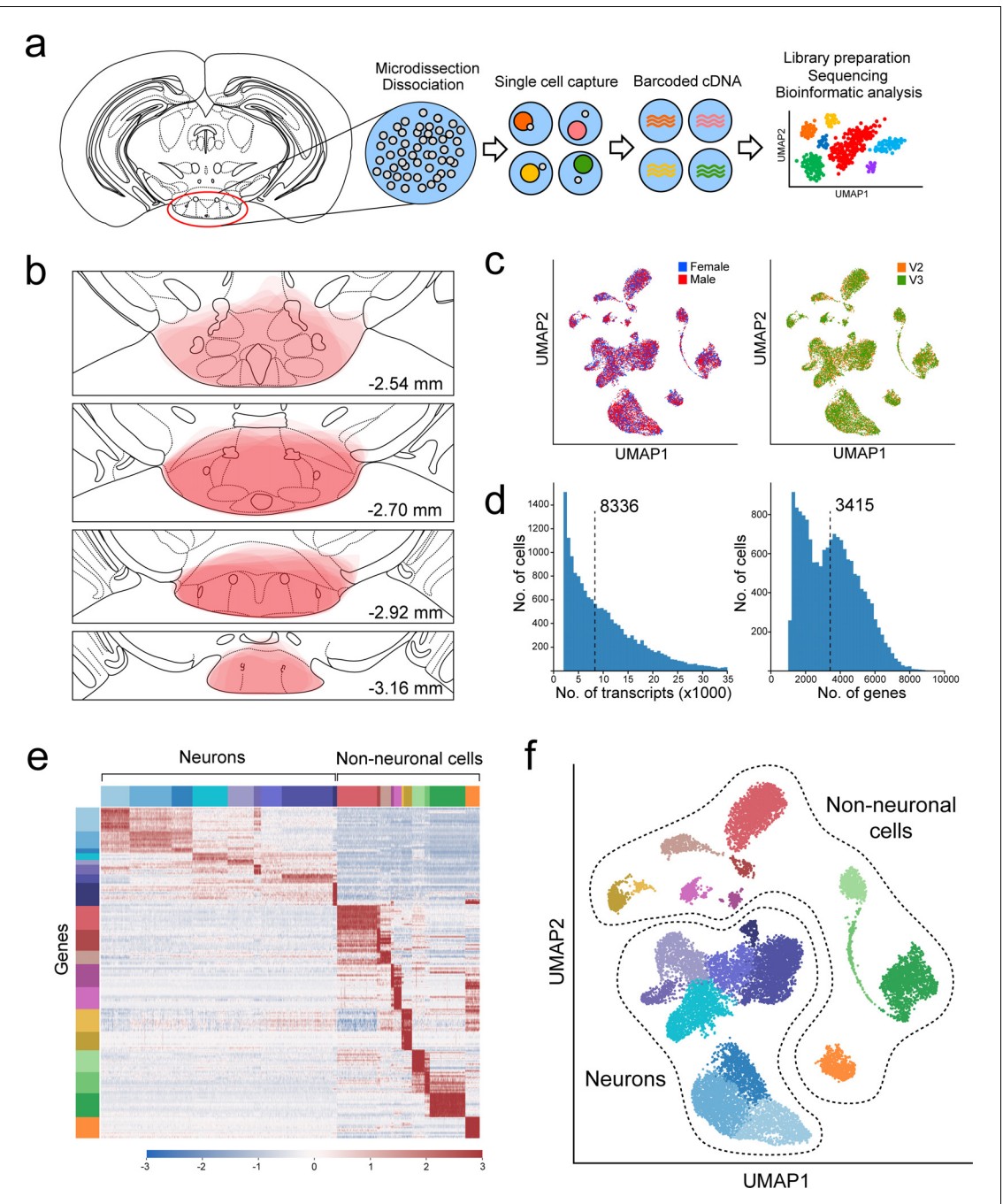

**Figure 1.** Overview of VPH microdissection, single-cell isolation, batch correction, and clustering. (a) Workflow schematic representing the VPH microdissection from coronal mouse brain slices, single-cell dissociation, sequencing library preparation, and bioinformatic analysis (*Mickelsen et al., 2019*). (b) Location of VPH microdissections mapped onto the coronal mouse brain atlas at distances from bregma of −2.54, −2.70, −2.92, and –3.16 mm. Atlas images were modified from *Paxinos, 2012*. (c) Two-dimensional UMAP plots representing 16,991 single cells from four sequencing libraries color-coded by mouse sex (left) and the 10x Genomics chemistry version (right) following batch correction. (d) Histograms of unique transcripts (left) and genes (right) were detected in 16,991 single cells after quality control. Dashed vertical lines represent the median transcripts and genes per cell, respectively. (e) Heatmap and (f) UMAP plot showing the first iteration of unsupervised clustering revealing 20 unique clusters. Neuronal populations are disjoint from non-neuronal populations.

The online version of this article includes the following figure supplement(s) for figure 1:

**Figure supplement 1.** Batch correction for sex and 10x Genomics chemistry versions.

**Figure supplement 2.** Proportion of cells derived from each sample and identification of discriminatory marker genes.

**Figure supplement 3.** Testing marker gene specificity with a classification model.

*Figure 1 continued on next page*

**Figure supplement 4.** Classification of VPH non-neuronal populations.

**Figure supplement 5.** Key discriminatory marker genes of oligodendrocyte-lineage cells, astrocytes, tanycytes, and ependymal cells.

number of genes per cell (*Figure 1d*), consistent with similar studies (e.g. *Mickelsen et al., 2019*; *Moffitt et al., 2018*; *Chen et al., 2017*), it is important to verify whether the marker gene expression presented is specific and robust. We first examined the distribution of transcripts captured per gene (*Figure 1—figure supplement 2e and f*) with the genes that we present as markers typically being the most highly expressed genes in the data set. Moreover, typical marker genes are expressed with at least 10 counts in a cell, and often with substantially more counts (*Figure 1—figure supplement 2g*).

We also verified that the markers presented are sufficient to uniquely describe each population. For this, we built a model that predicts the cluster identity of individual cells using only the expression of a small number of marker genes (*Figure 1—figure supplement 3a*). We then computed the number of incorrect cluster label predictions (as compared to each cells' original label as shown in *Figure 2b*). This computational experiment demonstrates that we can predict the neuronal cluster identity with over 90% accuracy with fewer than only 80 total genes (*Figure 1—figure supplement 3b–d*). Notably, this is less than 10% of the 1500 genes used to guide the neuronal dimensionality reduction and clustering analysis. Moreover, many of the misclassifications are between related clusters that share markers or within the clusters with uncertain cluster identity (ex. GLUT10, GLUT11, GABA12, GABA13; *Figure 1—figure supplement 3e*). Most clusters ascribed to discrete anatomical regions can be correctly labeled with almost 100% accuracy with fewer than five marker genes. This model suggests that the marker genes for the populations described in *Figures 3*, *4*, *5*, *6* and figure supplements are more than adequate to identify individual cells from these populations.

## Major non-neuronal cell types in the VPH

Among non-neuronal cell populations in the VPH, we identified 18 distinct clusters distinguished from one another by suites of cell type-specific discriminatory markers (*Figure 1—figure supplement 4a–d*). We resolved six distinct populations of oligodendrocyte lineage cells arranged in a contiguous strand in a UMAP plot (*Figure 1—figure supplement 4a*), likely reflecting a developmental gradient of gene expression. These include oligodendrocyte precursor cells (OPC or NG2+ cells; cluster 1), immature oligodendrocytes (cluster 2), and four oligodendrocyte populations (clusters 3–6), all of which exhibit a wave of differentially-expressed genes (*Cspg4*, *Fyn*, *Ctps*, *Tspans2*, *Apod*, *Klk6*, etc.; *Figure 1—figure supplement 5a and b*) that aligns well with the diversity of functional markers of the oligodendrocyte lineage in the mouse brain identified through previous scRNA-seq analyses (*Marques et al., 2016*; *Zeisel et al., 2018*; *Saunders et al., 2018*). We also resolved three distinct clusters of astrocytes (clusters 7, 8, and 9), all of which are *Aqp4*+ and *Agt*+. Clusters 7 and 8 are distinguishable by high expression of *Slc7a10* and *Htra1* and low expression of *Gfap*, while cluster 9 exhibits high expression of both *Gfap* and *C4*b (*Figure 1—figure supplement 5c and d*). In nearby clusters, we identified tanycytes (cluster 10: *Rax*+) and ependymal cells (cluster 11: *Ccdc153*+, *S100a6*+; *Figure 1—figure supplement 5c and d*). In addition, we found other distinct clusters readily identifiable as macrophages (cluster 12: *Mrc1*+), microglia (cluster 13: *Tmem119*+), pericytes (cluster 14: *Rgs5*+), vascular smooth muscle cells (VSMCs; cluster 15: *Rgs5*+, *Acta2*+), two populations of putative vascular leptomeningeal cells or VLMCs (cluster 16: *Fxyd5*+, *Slc47a1*+; cluster 17: *Dcn*+) and vascular endothelial cells (VECs; cluster 18: *Pecam1*+, *Slc38a5*+; *Figure 1—figure supplement 3b*). Our identification of major non-neuronal VPH cell types is based on, and closely aligns with, previous scRNA-seq analyses from mouse brain (*Marques et al., 2016*; *Zeisel et al., 2018*; *Saunders et al., 2018*).

## Diverse populations of excitatory and inhibitory neuronal cell clusters in the VPH

Among the neuronal clusters, which contained significantly more genes and UMIs per cell (4334 and 12,065, respectively) than the overall data set, we first examined broad patterns in the expression of fast amino acid and monoamine neurotransmitter markers (*Figure 2a*). Expression of genes

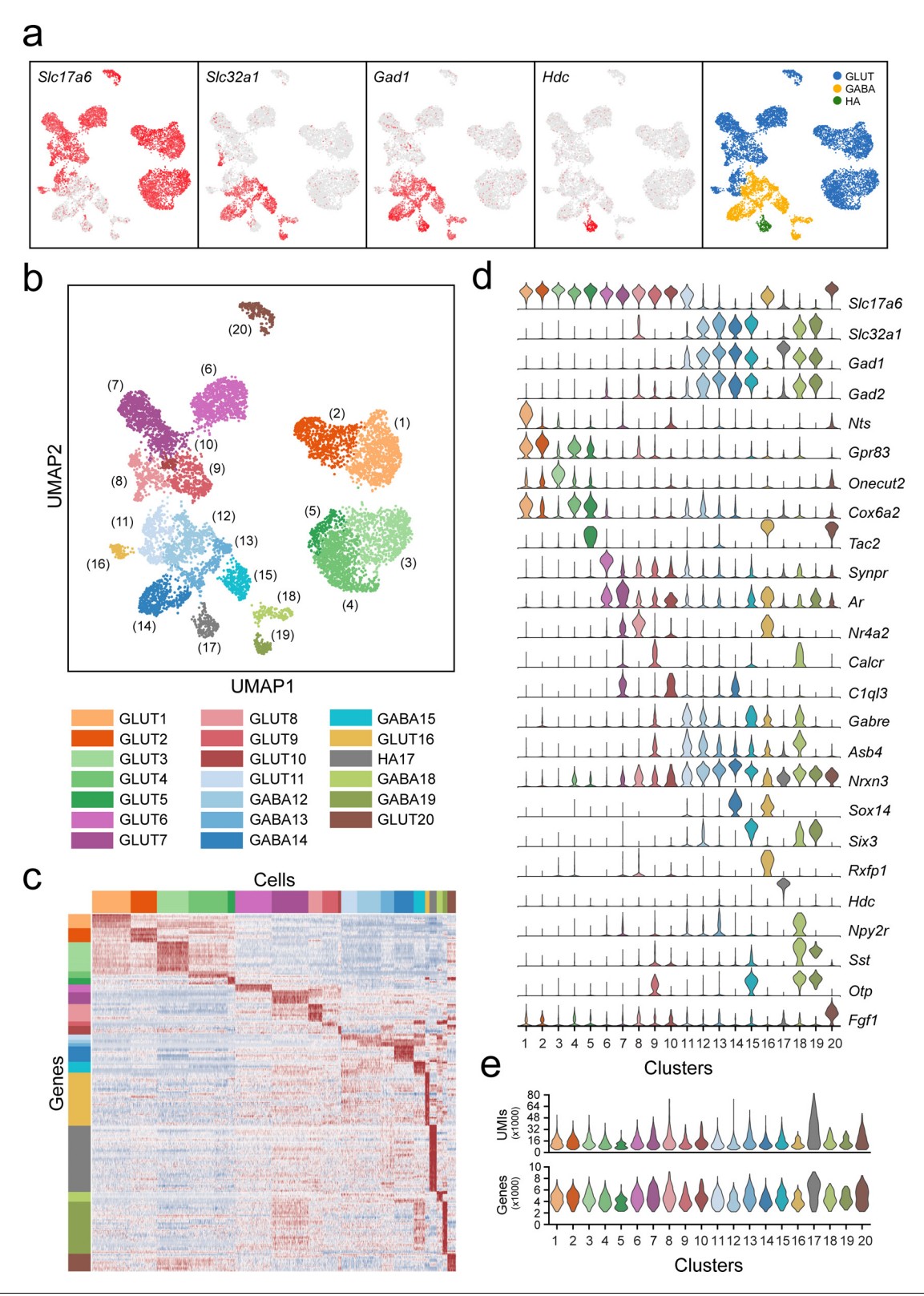

**Figure 2.** Classification of VPH neuronal populations. (a) UMAP plots showing normalized expression of *Slc17a6*, *Slc32a1*, *Gad1*, and *Hdc* after the second iteration of unsupervised clustering on just neuronal cells. Using these four genes, neurons were classified by a three-class Gaussian mixture model as glutamatergic (GLUT, blue), GABAergic (GABA, yellow), or histaminergic (HA, green). (b) Unsupervised clustering of 20 VPH neuronal cell types shown in a UMAP embedding. (c) Heatmap showing scaled expression of discriminatory genes across all 20 neuronal clusters. (d) Violin plot
*Figure 2 continued on next page*

*Figure 2 continued*

showing the distribution of normalized expression in each cluster of neurotransmitters (*Slc17a6*, *Slc32a1*, *Gad1*, *Gad2*) (upper) and discriminatory marker genes (lower). (e) Violin plots showing the distribution of the number of unique transcripts (upper) and the number of genes (lower) in each neuronal cluster.

necessary for the synthesis and packaging of the excitatory transmitter glutamate (*Slc17a6*, encoding the vesicular glutamate transporter 2, VGLUT2) and the inhibitory transmitter GABA (*Slc32a1*, encoding the vesicular GABA transporter, VGAT) provided a binary classification of *Slc17a6*+ clusters as glutamatergic (VPH^GLUT) and *Slc32a1*+ clusters as GABAergic (VPH^GABA) neurons. This was further supported by the expression of the gene encoding a synthetic enzyme for GABA (*Gad1*/GAD67) which largely aligns with *Slc32a1*+ clusters. We found that of the 20 neuronal clusters we identified, thirteen are glutamatergic, six are GABAergic and one (cluster 17) best matched the profile of a unique population of histaminergic (HA) neurons based on the unique expression of histidine decarboxylase (*Hdc*/HDC; *Figure 2a,b*). Overall, within these neuronal populations, clusters are distinguished by suites of differentially expressed transcripts (*Figure 2c,d*) with comparable UMIs/cell and genes/cell (*Figure 2e*). In the following analyses, we validated the co-expression of key markers and their spatial organization in selected VPH neuronal populations. In all cases, we first attempted to map transcriptionally distinct cell clusters onto specific anatomical subregions within the VPH, by comparing differentially-expressed transcripts with the online database of in situ hybridization (ISH) data from the Allen Brain Atlas (ABA; *Lein et al., 2007*), followed by co-expression analysis using multiplexed FISH.

## Transcriptional signatures differentiate ventral and dorsal premammillary (PM) nuclei

Among neuronal clusters that correspond to discrete anatomical subregions of the VPH, we identified at least one VPH^GLUT population with a suite of differentially-expressed transcripts that appears to closely match the ventral premammillary nucleus (PMv; *Figure 3*). VPH^GLUT cluster 7 is enriched in the following key transcripts: *Tac1* (encoding substance P or SP), *Nos1* (encoding neuronal nitric oxide synthase or NOS), *Calb2* (encoding calretinin), and *Foxp2* (encoding the transcription factor forkhead box P2; *Figure 3a*) along with a host of other differentially expressed markers (*Figure 3b*). The spatial pattern of expression of each of the markers in *Figure 3a* were found in the ABA (*Lein et al., 2007*), and correspond closely to the PMv (*Figure 3c*). Through FISH co-expression analysis, we found that *Tac1*, *Calb2*, and *Slc17a6* are extensively co-expressed in a large cluster of neurons in the PMv (*Figure 3d*). These data broadly align with several known markers for the PMv (*Donato and Elias, 2011*). For example, SP (*Shimada et al., 1988*; *Larsen, 1992*) and NOS (*Vincent and Kimura, 1992*) are both enriched in the PMv. Another marker that is enriched in VPH^GLUT cluster 7 is *Adcyap1* (encoding the neuropeptide pituitary adenylate cyclase-activating polypeptide, or PACAP; *Figure 3b*). Consistent with the role of the PMv in reproductive function (*Donato and Elias, 2011*; *Leshan and Pfaff, 2014*), PACAP+ PMv neurons were recently shown to critically regulate female reproductive physiology and fertility (*Ross et al., 2018*).

Interestingly, we also found that a distinct subpopulation of VPH^GLUT cluster 7 neurons robustly expresses markers of catecholaminergic neurotransmission including *Slc6a3* (encoding the dopamine transporter, DAT), *Ddc* (encoding DOPA decarboxylase), and *Slc18a2* (encoding the vesicular monoamine transporter 2, VMAT2) but low expression of *Th* (encoding tyrosine hydroxylase; *Figure 3e*). This cluster corresponds well to a previously identified catcholaminergic (*Zoli et al., 1993*), *Slc6a3*+ PMv population (*Meister and Elde, 1993*) that was more recently found, through circuit and behavioral analyses, to regulate male social behavior (*Soden et al., 2016*) and aggression (*Stagkourakis et al., 2018*) in a glutamate-dependent, but dopamine-independent, manner (*Soden et al., 2016*), consistent with the profile we identified (*Slc17a6*+, *Slc18a2*+, *Slc6a3*+, *Ddc*+, *Th*-). To probe further PMv subpopulations, we subjected VPH^GLUT cluster 7 to another iteration of clustering and found that it could be parsed into eight subclusters (*Figure 3—figure supplement 1a*). Each of the eight subclusters expressed *Slc17a6* but exhibit differential expression of major markers such as *Tac1* and *Foxp2* (*Figure 3—figure supplement 1b–d*). Notably, subclustering revealed two distinct populations of putative catecholaminergic *Tac1*+/*Foxp2*+ neurons (clusters 7–1 and −2), both of which express *Slc18a2* but only one of which expresses *Slc6a3*. Taken together,

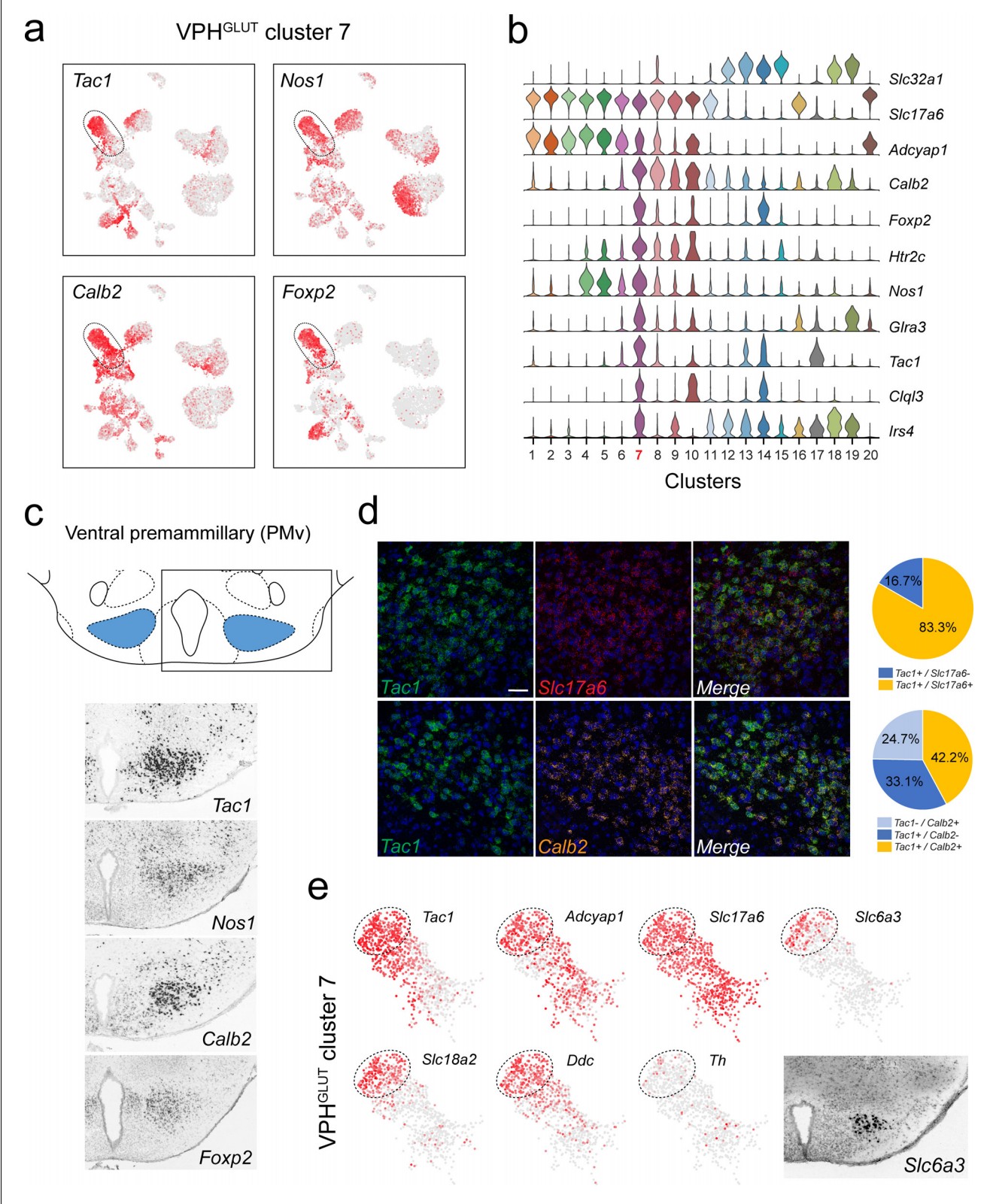

**Figure 3.** Identification of a population of putative PMv neurons and a catecholaminergic PMv subpopulation. (a) UMAP plots showing normalized expression of *Tac1*, *Nos1*, *Calb2*, and *Foxp2* enriched in VPH^GLUT cluster 7 following *Slc32a1* and *Slc17a6* (top). (b) Violin plot showing discriminatory marker genes enriched in cluster 7. (c) Mouse brain atlas schematic, modified from *Paxinos, 2012*, showing the PMv in a coronal section at distance from bregma of −2.46 mm (top). ISH images for *Tac1*, *Nos1*, *Calb2*, and *Foxp2* from the ABA (*Lein et al., 2007*; bottom). (d) Confocal micrographs

*Figure 3 continued on next page*

*Figure 3 continued*

(40×) of FISH in coronal sections of wild type mice and corresponding pie charts representing co-expression of mRNA for *Tac1* and *Slc17a6* (n = 678 cells, three mice; upper) and *Tac1* and *Calb2* (n = 963 cells, three mice; lower). Scale bar (applicable to all micrographs) 50 μm. (e) UMAP plots showing normalized expression of markers in VPH^GLUT cluster 7 only, including cell type markers for reference (*Tac1, Adcyap1,* and *Slc17a6*) and markers that define a subpopulation of putative catecholaminergic neurons (*Slc6a3, Slc18a2* and *Ddc,* but with very low *Th*). ISH image from the ABA (***Lein et al., 2007***) showing *Slc6a3* expression in the PMv (inset).

The online version of this article includes the following figure supplement(s) for figure 3:

**Figure supplement 1.** Subclustering of PMv neurons (VPH^GLUT cluster 7).

these transcriptomic data provide further biological insight into the repertoire of receptors and signaling molecules expressed by this key behavioral node.

Another distinct VPH^GLUT population (cluster 6) share a number of common markers with clusters 1–5, for example *Cck* (encoding the neuropeptide cholecystokinin) and *Foxb1* (encoding the transcription factor Forkhead Box B1), which are largely undetectable in clusters 7–10 (***Figure 4a,b***). Curiously, VPH^GLUT cluster 6 also expresses a suite of markers that are enriched in clusters 7–10 (ex. *Ebf3, Dlk1, Synpr, Nxph1*) but largely undetectable in clusters 1–5 (***Figure 4a,b***). In particular, *Synpr* (encoding the presynaptic protein synaptoporin), *Dlk1* (encoding delta like non-canonical Notch ligand 1), *Ebf3* (encoding early B cell factor 3) and *Stxbp6* (encoding the synaptic protein syntaxin binding protein 6) are enriched in VPH^GLUT cluster 6 (***Figure 4a,b***). Examining the expression patterns of *Synpr, Dlk1,* and *Stxbp6* in the ABA (***Lein et al., 2007***), we found that all three are enriched in a discrete region that appears to correspond well to the dorsal PM (PMd; ***Figure 4c,d***) and are largely undetectable in the more caudal mammillary bodies (MB; ***Figure 4d***). This suggests that VPH^GLUT cluster 6 likely represents a PMd population that expresses a number of unique signatures (ex. *Synpr, Stxbp6*) but shares some commonalities with VPH^GLUT clusters 1–5 (ex. *Foxb1, Cck, Adcyap1*) and VPH^GLUT cluster 7, a putative PMv population (ex. *Nxph1, Ar*). These data suggest that the spatially segregated PMd and PMv may be defined by distinct transcriptional signatures.

## A neuronal population in the supramammillary (SUM) nuclei

Another notable neuronal cluster is VPH^GLUT cluster 8. Subjecting this cluster to another iteration of unsupervised clustering revealed six subclusters (***Figure 4—figure supplement 1a***) which exhibit suites of differentially expressed genes (***Figure 4—figure supplement 1b,c***). Several of the markers that define distinct subclusters within VPH^GLUT cluster 8 are found within anatomically identified neuronal populations in the SUM (***Figure 4—figure supplement 1d***), only the most ventral portion of which would be included in our microdissection (***Figure 1b***). These markers, previously identified in rodents include *Th* (encoding tyrosine hydroxylase, enriched in cluster 8–1; ***Swanson, 1982***) and *Nos1* (enriched in cluster 8–5; ***Yamada et al., 1996***; ***Pedersen et al., 2017***). Although all cells in cluster 8 express *Slc17a6*, and are nominally classified as glutamatergic, at least one subpopulation appears to co-express *Slc32a1* and *Gad2* but not *Gad1* (***Figure 4—figure supplement 1e***). Subcluster 8–3 closely corresponds to this *Slc17a6+/Slc32a1+/Gad2+* subcluster and co-expresses a suite of discriminatory makers including *Sema3c, Inhba, Rxfp1* and others (***Figure 4—figure supplement 1f***). Cross-referencing with the ABA (***Lein et al., 2007***) shows selective expression of *Sem3c, Inhba,* and *Rxfp1* in the SUM (***Figure 4—figure supplement 1f***). Another subcluster, cluster 8–4 also expresses a moderate level of *Slc32a1* (***Figure 4—figure supplement 1c***). Interestingly, a unique population of VGLUT2/VGAT co-expressing axons originating in the SUM has been described among projections to the hippocampal dentate gyrus (***Boulland et al., 2009***; ***Soussi et al., 2010***) and these dual phenotype SUM neurons were found to indeed co-release GABA and glutamate onto neurons of the dentate gyrus (***Pedersen et al., 2017***; ***Hashimotodani et al., 2018***). Our identification of this VPH^GLUT cluster 8 subpopulation is consistent with previous anatomical and functional data defining a unique dentate-projecting, dual phenotype SUM population but would require additional functional evidence to validate. However, the markers identified in our transcriptomic analysis may provide possible strategies for their precise genetic targeting. Furthermore, a microdissection that includes the entirety of the SUM would provide a more comprehensive picture of neuronal cell type diversity within this region.

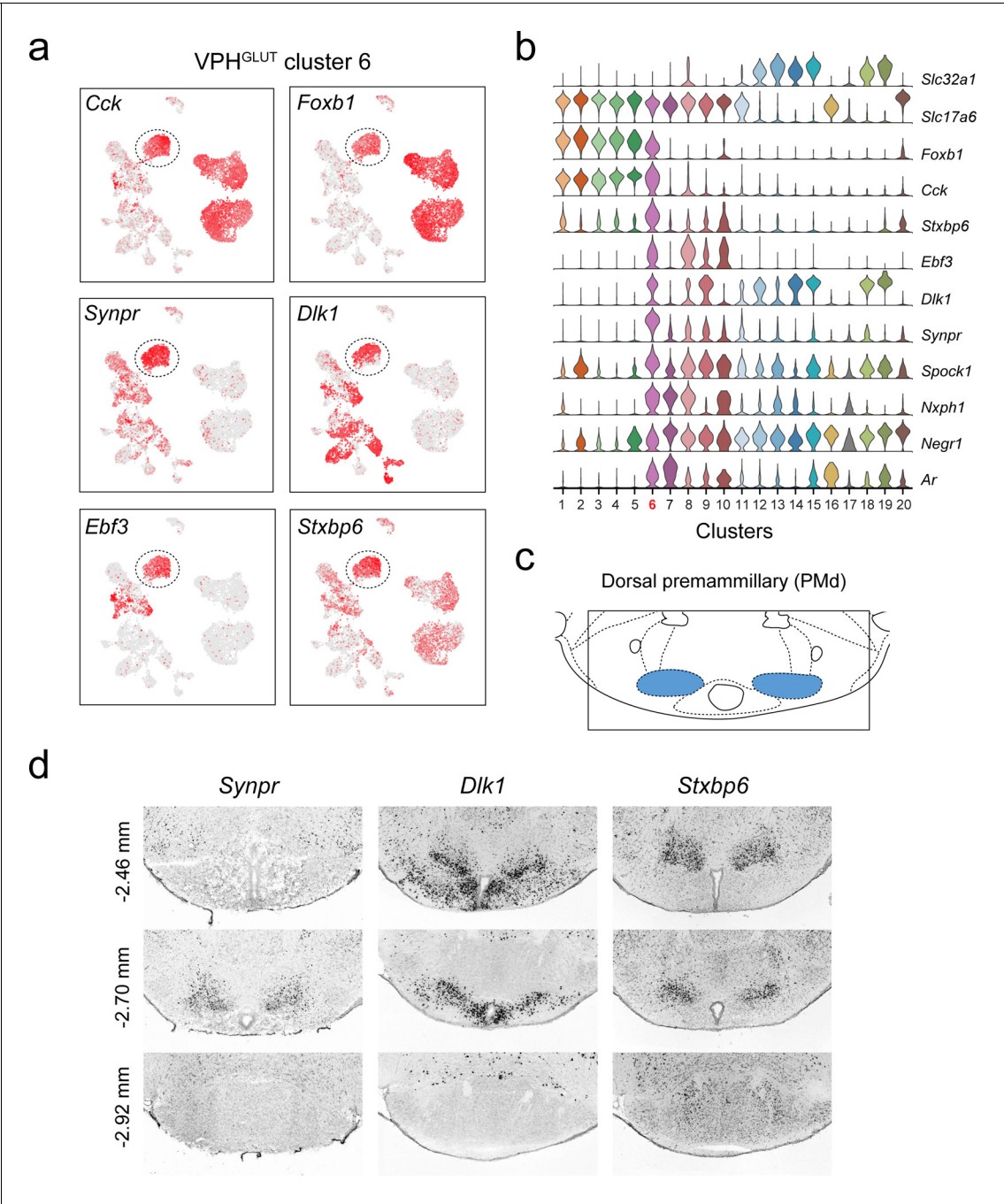

**Figure 4.** Identification of a population of putative PMd neurons (VPH^GLUT cluster 6). (**a**) UMAP plots showing normalized expression of *Cck*, *Foxb1* (shared with VPH^GLUT clusters 1–5), *Synpr*, *Dlk1*, *Ebf3*, and *Stxbp6* (enriched in VPH^GLUT cluster 6). (**b**) Violin plot showing discriminatory marker genes enriched in VPH^GLUT cluster 6 following *Slc32a1* and *Slc17a6* (top). (**c**) Mouse brain atlas schematic (*Paxinos, 2012*) showing the PMd in a coronal section at distance from bregma of −2.70 mm (top). (**d**) ISH images for three anterior to posterior coronal sections (approximate distance from bregma −2.46,–2.80, and −2.92 mm) for *Synpr* (left), *Dlk1* (middle), and *Stxbp6* (right) from the ABA (*Lein et al., 2007*; bottom). In each case, expression appears to be enriched in the PMd in anterior sections and largely absent in the MB in the posterior section.

The online version of this article includes the following figure supplement(s) for figure 4:

**Figure supplement 1.** Subclustering of putative SUM neurons (VPH^GLUT cluster 8).

**Figure supplement 2.** Identification of multiple populations of putative Arc neurons (VPH^GABA clusters 14, 18, and VPH^GLUT cluster 16).

**Figure supplement 3.** Identification and subclustering of multiple populations of putative LHA/Tub neurons (VPH^GABA clusters 15 and 19).

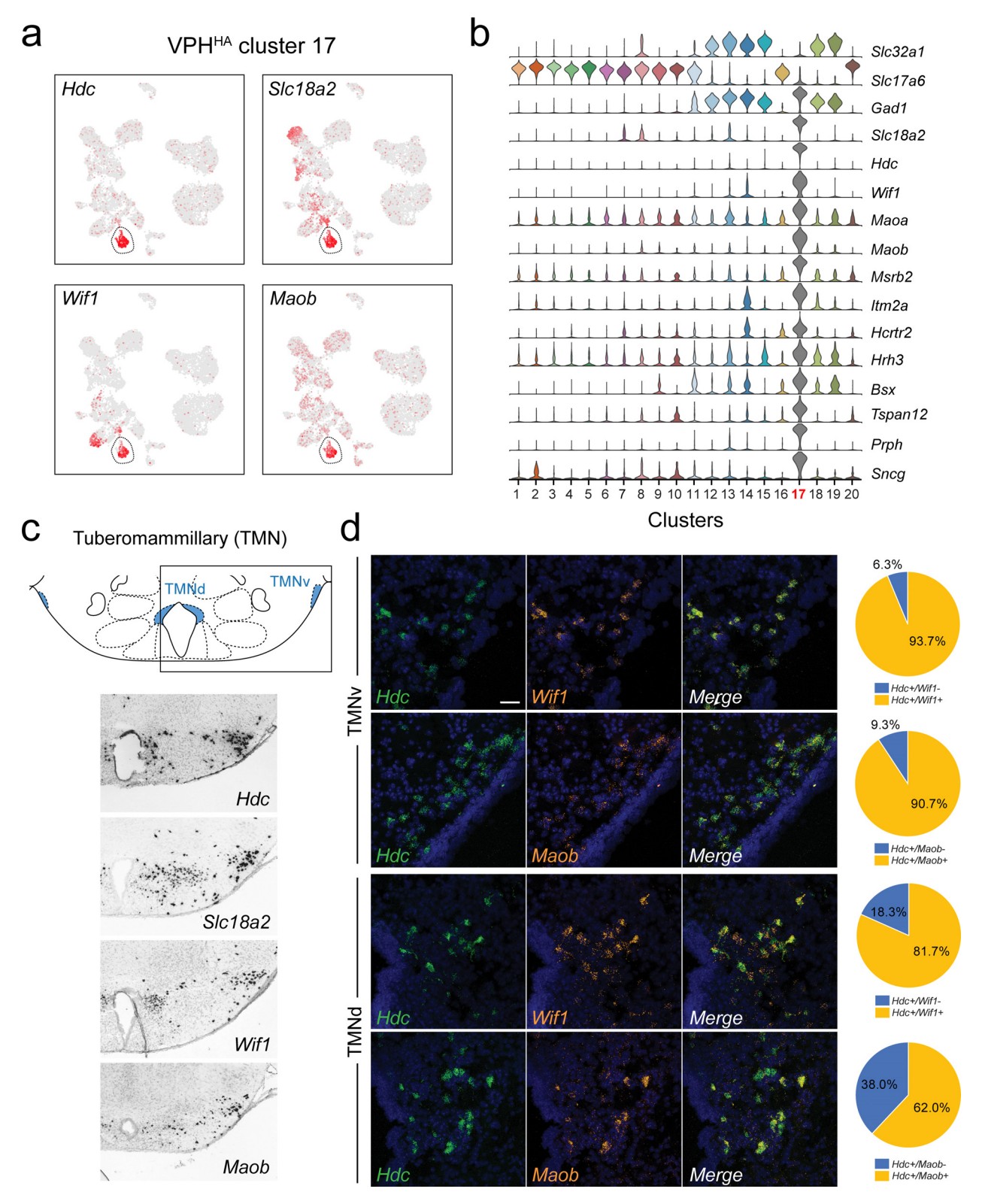

**Figure 5.** Identification of histaminergic (HA) neurons in the TMN. (**a**) UMAP plots showing normalized expression of *Hdc*, *Slc18a2*, *Wif1*, and *Maob* enriched in VPH^HA cluster 17 (circled). (**b**) Violin plot showing discriminatory marker genes enriched in VPH^HA cluster 17 following *Slc32a1* and *Slc17a6* (top). (**c**) Mouse brain atlas schematic, modified from *Paxinos, 2012*, showing the dorsal TMN (TMNd) and ventral TMN (TMNv) in a coronal section at distance from bregma of −2.54 mm (top). ISH images for *Hdc*, *Slc18a2*, *Wif1*, and *Maob* from the ABA (*Lein et al., 2007*; bottom). (**d**) Confocal
*Figure 5 continued on next page*

Figure 5 continued

micrographs (40×) of FISH in coronal sections of wild type mice and corresponding pie charts representing co-expression of mRNA in the TMNv (top) for *Hdc* and *Wif1* (n = 415 cells, three mice; upper) and *Hdc* and *Maob* (n = 570 cells, three mice; lower) and in the TMNd (bottom) for *Hdc* and *Wif1* (n = 109 cells, three mice; upper) and *Hdc* and *Maob* (n = 192 cells, three mice; lower). Scale bar (applicable to all micrographs) 50 μm.

The online version of this article includes the following figure supplement(s) for figure 5:

**Figure supplement 1.** Subclustering of TMN HA neurons (VPH$^{HA}$ cluster 17) and a population of HA-like neurons (subcluster of VPH$^{GABA}$ cluster 13).

## Neuronal populations in the caudal arcuate (Arc) and lateral hypothalamic/tuberal (LHA/Tub) nuclei

A number of neuronal clusters are defined by markers known to be enriched in the hypothalamic Arc nucleus, despite our microdissection likely capturing only the most caudal portion of the Arc. Clusters that express Arc-enriched markers (*Romanov et al., 2017*; *Campbell et al., 2017*), including cluster 14 (*Unc13c*, *Trh*, etc.), cluster 16 (*Tac2*, *Pdyn*, etc.), and cluster 18 (*Agrp*, *Sst*, etc.) were combined in a second iteration of clustering to reveal eight subclusters defined by differentially expressed genes (*Figure 4—figure supplement 2a–c*). Cluster 14 parsed into five subclusters with markers that include (*Unc13c*, *Thrb* and *Trh*; *Figure 4—figure supplement 2e*), that appear to correspond well to a population of cells in the caudal Arc. In particular, using ISH data from the ABA (*Lein et al., 2007*), both *Dlk1* and *Thrb* (encoding the thyroid hormone receptor beta) are enriched in a cluster surrounding the 3rd ventricle/mammillary recess corresponding to the medial posterior portion of the Arc (*Paxinos, 2012*) or posterior portion of the periventricular nucleus (Allen Institute Mouse Brain Reference Atlas). Cluster 18 parsed into two subclusters. While both express *Otp*, one subcluster likely corresponds to well-known Arc AGRP/NPY neurons (*Agrp* and *Npy*), and another likely corresponds to a previously identified population of Arc *Sst*+ neurons such as *Sst*/*Unc13c* (*Campbell et al., 2017*).

A highly distinct neuronal cluster, VPH$^{GLUT}$ cluster 16, which did not subcluster further, expresses a suite of markers that identify it as a well-described neuronal cell type in the Arc - kisspeptin-neurokinin B-dynorphin (KNDy) neurons (*Figure 4—figure supplement 2f*). KNDy neurons are considered to be the gonadotropin-releasing hormone pulse generator controlling release of luteinizing hormone from the anterior pituitary, and therefore essential for fertility and reproduction (*Lehman et al., 2010*; *Moore et al., 2018*; *Harter et al., 2018*). We confirm that this cluster co-expresses the defining neuropeptides *Tac2* (encoding neurokinin B) and *Pdyn* (encoding dynorphin; *Figure 4—figure supplement 2c,f*). The key markers that define KNDy neurons in our data set are consistent with those identified in previous mouse scRNA-seq data sets, both from broader hypothalamic samples (*Chen et al., 2017*; *Romanov et al., 2017*) and Arc-specific samples (*Campbell et al., 2017*). Consistent with the important role of Arc KNDy neurons in reproduction and fertility, we found high expression of the following hormone receptors: *Prlr* (encoding the prolactin receptor), *Esr1* (encoding the estrogen receptor 1), *Ar* (encoding the androgen receptor), and *Pgr* (encoding the progesterone receptor). Other markers that exhibit relatively robust and unique expression in KNDy neurons include *Nhlh2*, *Inhbb*, *Nr5a2*, *Rxfp1* and *Col2a1*. A selection of key markers for VPH$^{GLUT}$ cluster 16 neurons (*Tac2*, *Pdyn*, and *Esr1*) were confirmed using ISH data from the ABA (*Lein et al., 2007*; *Figure 4—figure supplement 2f*) and correspond to the caudal Arc.

Finally, two other distinct GABAergic neuronal populations (VPH$^{GABA}$ clusters 15 and 19; *Figure 4—figure supplement 3a*) appear to correspond to the tuberal region of the LHA, also known as the hypothalamic tuberal nuclei (*Figure 4—figure supplement 3d*). The most caudal part of this region would likely be included in the rostral-most slices in our microdissection. VPH$^{GABA}$ cluster 15 (enriched in *Sst*, *Six3*, *Otp*, *Col25a1*, *Ecel1*, and *Parm1*), which may be parsed into four subclusters, and VPH$^{GABA}$ cluster 19 (enriched in *Sst*, *Otp*, *Dlk1*, *Pthlh*, and *Ptk2b*; *Figure 4—figure supplement 3b,c,e*), correspond closely with two previously identified subpopulations of *Sst*-expressing GABAergic neurons in the tuberal LHA (LHA$^{GABA}$ clusters 6 and 13 , respectively; *Mickelsen et al., 2019*). *Sst*-expressing neurons are enriched in the tuberal region (*Morales-Delgado, 2011*) and have recently been implicated in the regulation of feeding behavior in mice (*Luo et al., 2018*).

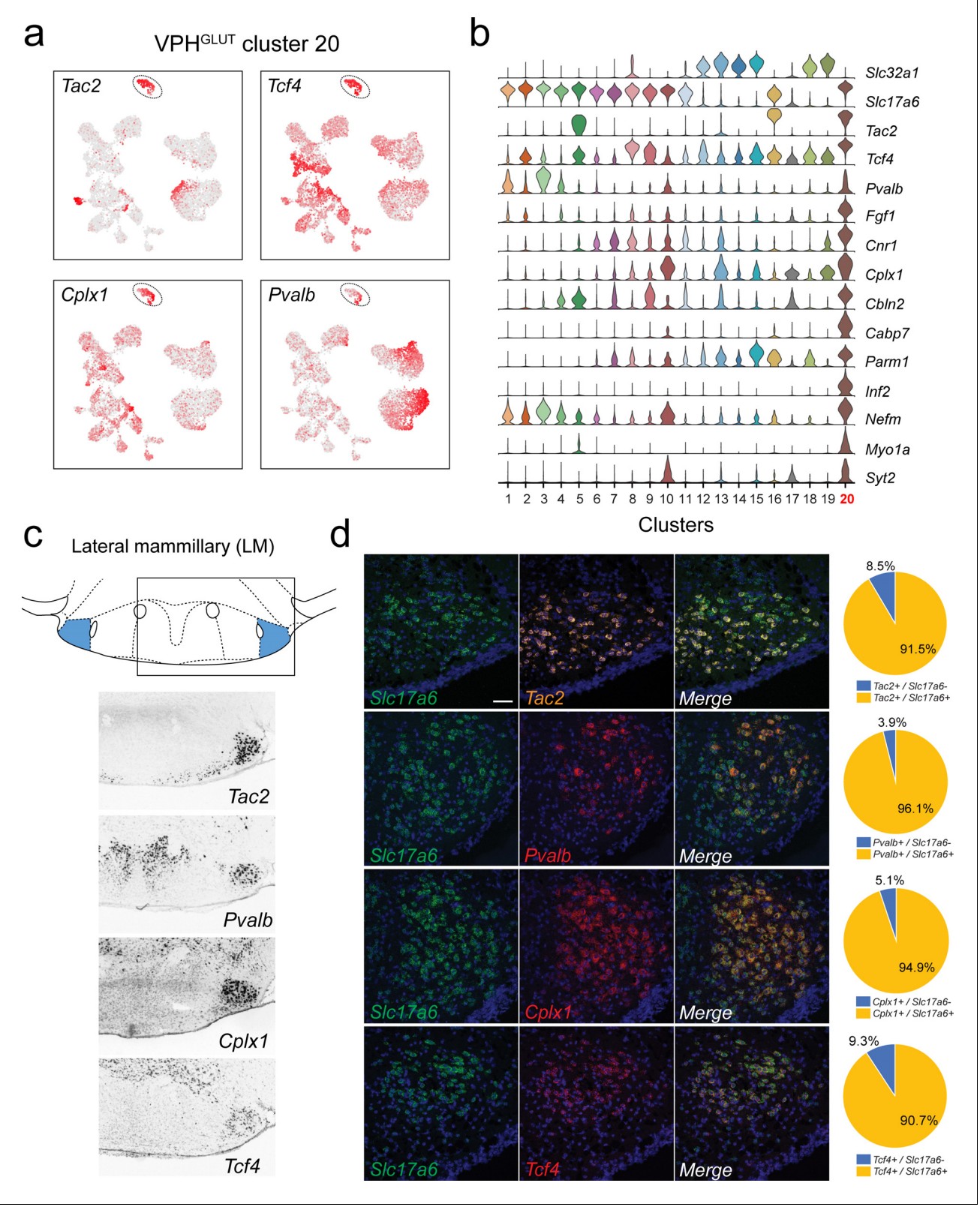

**Figure 6.** Identification of lateral mammillary (LM) neurons. (**a**) UMAP plots showing normalized expression of *Tac2*, *Tcf4*, *Cplx1,* and *Pvalb* enriched in VPH[GLUT] cluster 20 (circled). (**b**) Violin plot showing discriminatory marker genes enriched in VPH[GLUT] cluster 20 following *Slc32a1* and *Slc17a6* (top). (**c**) Mouse brain atlas schematic, modified from *Paxinos, 2012*, showing the LM in a coronal section at distance from bregma of −2.92 mm (top). ISH images for *Tac2*, *Tcf4*, *Cplx1,* and *Pvalb* from the ABA (*Lein et al., 2007*; bottom). (**d**) Confocal micrographs (40×) of FISH in coronal sections of wild

*Figure 6 continued on next page*

*Figure 6 continued*

type mice and corresponding pie charts representing co-expression of mRNA for *Slc17a6* and *Tac2* (n = 1210 cells, three mice; upper), *Slc17a6* and *Pvalb* (n = 1057 cells, three mice, upper middle), *Slc17a6* and *Cplx1* (n = 1269 cells, three mice; lower middle), and *Slc17a6* and *Tcf4* (n = 1642 cells, three mice; lower), Scale bar (applicable to all micrographs) 50 µm.

The online version of this article includes the following figure supplement(s) for figure 6:

**Figure supplement 1.** Subclustering of LM neurons (VPH^GLUT cluster 20).

## Histaminergic neurons in the tuberomammillary (TMN) nuclei

A highly distinct cluster emerged from our data set that exhibits a pattern of gene expression characteristic of histamine (HA)-producing neurons (VPH^HA cluster 17). HA neurons are a well-characterized monoaminergic, neuromodulatory population with cell bodies concentrated in the ventral TMN (TMNv) and dorsal TMN (TMNd), and also scattered throughout the VPH region. TMN HA neurons are the sole source of neuronally-synthesized HA in the brain, make widespread projections throughout the brain (*Ericson et al., 1987*; *Inagaki et al., 1988*) and have a key role in modulating wakefulness (*Brown et al., 2001*; *Haas et al., 2008*; *Panula and Nuutinen, 2013*). Previous mouse hypothalamic scRNA-seq studies have also detected a population of HA neurons (*Chen et al., 2017*; *Campbell et al., 2017*; *Kim et al., 2020*). We found that VPH^HA cluster 17 neurons express the following key transcripts: *Hdc*, in addition to *Slc18a2* (encoding the vesicular monoamine transporter two or VMAT2), *Maob* (encoding monoamine oxidase B), and another unique marker Wif1 (encoding Wnt inhibitory factor 1; *Figure 5a*). Additional makers include transcripts for *Maoa*, *Msrb2*, *Itm2a*, *Bsx*, *Hrh3*, *Hcrtr2*, *Sncg*, and *Prph* (*Figure 5b*). Our identification of *Prph* (encoding the neurofilament protein peripherin) as a highly selective marker for HA neurons is consistent with previous anatomical work (*Eriksson et al., 2008*). Key markers (*Hdc*, *Slc18a2*, *Wif1*, and *Maob*) correspond well to the TMNv and TMNd in the ABA (*Lein et al., 2007*; *Figure 5c*). FISH analysis of *Wif1* and *Maob* co-expression with *Hdc* revealed extensive co-expression in both the TMNv and TMNd but with lower coexpression in the TMNd (*Figure 5d*). With regard to fast neurotransmitter phenotype, we found that VPH^HA cluster 17 neurons are exceptional in our data set in that they express very low levels of both *Slc17a6* and *Slc32a1*, but robustly express *Gad1*. That TMN HA neurons are GAD+, and likely capable of GABA synthesis, has long been recognized (*Vincent et al., 1983*; *Takeda et al., 1984*). While there has been recent conflicting data on whether or not HA neurons express *Slc32a1* (*Yu et al., 2015*; *Venner et al., 2019*), our data indicate that TMN HA neurons robustly co-express *Gad1* accompanied by very low co-expression of *Slc32a1*.

To investigate the possibility that TMN HA neurons are themselves heterogeneous, we subjected VPH^HA cluster 17 neurons to another iteration of clustering. However, we also observed that a subset of VPH^GABA cluster 13 neurons share some common markers with HA neurons and thus included both clusters 13 and 17 in our subclustering analysis. VPH^HA cluster 17 parsed into four subclusters (subclusters 7–10; *Figure 5—figure supplement 1a*). While cluster 13 parsed into six subclusters (subclusters 1–6), only subcluster 5, which we refer to as HA-like, share some key features with subclusters 7–10, collectively denoted by the perforated outline (*Figure 5—figure supplement 1a*). Subclusters 9 and 10 comprise the majority of cluster 17 neurons and express all the cardinal signatures of HA neurons (*Hdc*, *Slc18a2*, *Prph*, *Wif1*, *Hcrtr2*, *Hrh3*). Subclusters 7 and 8 are much smaller populations that express variable levels of many of these same markers while also exhibiting highly unique markers (ex. *Cckar*, *Wnt7a*, *Gpc5*, and *Rspo2*; *Figure 5—figure supplement 1c,d*). Interestingly, HA-like subcluster five is unique in expressing a number of HA markers including *Hdc* (but at low levels), *Slc18a2*, *Prph*, *Wif1*, and *Hrh3* in addition to several more unique makers such as *GM39653*, *Npy2r*, and *Chodl* (*Figure 5—figure supplement 1c,d*). Importantly, subcluster five is a putative GABAergic population with moderate co-expression of *Slc32a1*. Taken together, these data suggest a degree of transcriptional heterogeneity among HA neurons, which requires a larger, and perhaps enriched, sampling of *Hdc*+ neurons to fully resolve.

## A distinct neuronal cluster corresponds to the lateral mammillary (LM) nuclei

We identified a distinct VPH^GLUT population (cluster 20) characterized by the expression of *Tac2*, *Pvalb* (encoding the calcium-binding protein parvalbumin), *Cplx1* (encoding the synaptic protein

complexin-1), and *Tcf4* (encoding transcription factor 4; ***Figure 6a***), in addition to a suite of other discriminatory markers including *Fgf1, Cbln2, Infg2, Nefm, Myo1a,* and *Syt2* (***Figure 6b***). The spatial pattern of expression of *Tac2, Pvalb, Cplx1,* and *Tcf4* in the ABA (***Lein et al., 2007***) corresponds well to the lateral mammillary (LM) nuclei, which are tightly circumscribed bilateral cell clusters, immediately lateral to the medial mammillary region and bisected by the fornix (***Figure 6c***). LM neurons are components of the MB and are implicated in the encoding of head direction and aspects of spatial memory (***Vann and Aggleton, 2004***; ***Dillingham et al., 2015***; ***Dillingham and Vann, 2019***). Consistent with our scRNA-seq results, FISH analysis revealed extensive co-expression of *Tac2, Pvalb, Cplx1,* and *Tcf4* with *Slc17a6* in the LM. Notably, *Pvalb* expression is limited to a subset of *Slc17a6+* neurons in the LM (***Figure 6d***). These data suggest that LM neurons comprise a transcriptionally-distinct population that appears to be confined to the anatomical boundaries of the LM.

Although VPH^GLUT^ cluster 20 is highly distinct relative to other clusters in UMAP space (***Figure 2b***), we found that it is not entirely homogenous. While *Tcf4* is distributed evenly throughout cluster 20, one pole is enriched in *Pvalb* (***Figure 6a***). To explore the possibility of LM heterogeneity, we subjected cluster 20 to another round of clustering resulting in three distinct subclusters (***Figure 6—figure supplement 1a***), each defined by a number of discriminatory markers (***Figure 6—figure supplement 1b,c,d***). For example, *Pvalb* and *Col11a1* are enriched in subcluster 1, while *Fxyd6* and *Slc7a3* are enriched in subcluster 3, suggesting a degree of cell type heterogeneity among LM neurons.

## Multiple neuronal populations correspond to the medial mammillary region

We identified five VPH^GLUT^ clusters that appear to be both closely interrelated to each other transcriptionally and distinct from other neuronal clusters. VPH^GLUT^ clusters 1–5 are arranged in close proximity in UMAP space (***Figure 2b***), with clusters 1 and 2 forming one contiguous cluster and clusters 3–5 forming another. Clusters 1–5 share a number of common markers including genes that encode the neuropeptides *Cartpt, Cck, Adcyap1* as well as the transcription factor *Foxb1* (***Figure 7a,b***). Cross-referencing *Cartpt, Foxb1, Cck,* and *Adcyap1* with the ABA (***Lein et al., 2007***) indicate that VPH^GLUT^ clusters 1–5 correspond well to the medial mammillary region (***Figure 7c***), which itself may be subdivided into cytoarchitecturally-defined anatomical compartments: median (MnM), medial (MM), and lateral (ML) subdivisions (***Paxinos, 2012***). Notably, *Cartpt, Foxb1,* and *Cck,* show low expression in VPH^GLUT^ cluster 20 (LM) in our scRNA-seq data and correspondingly low expression in the LM in ABA ISH data (***Figure 7c***). In contrast, *Adcyap1* shows high expression in cluster 20 and high expression in the LM in ISH data, further reinforcing the notion that clusters 1–5 represent the medial mammillary region, and cluster 20 represents the LM. These data are consistent with other scRNA-seq analyses of whole mouse hypothalamus, which revealed a single population of *Foxb1+* neurons ascribed to the mammillary nuclei in adult mice (***Chen et al., 2017***) and most recently, several in the developing mouse hypothalamus (***Kim et al., 2020***). Multiplex FISH analysis revealed extensive co-expression of the neuropeptide transcripts *Cck, Adcyap1,* and *Cartpt* with *Slc17a6* in the region (***Figure 7d***). Other discriminatory markers that we found are common to clusters 1–5 include *Rprm, Cpne9, Ctxn3, Fam19a1,* and *Gpr83* (***Figure 7b*** and ***Figure 7—figure supplement 1a***). Notably, we found that *Cpne9* (encoding copine-9) is uniquely enriched in clusters 1–5, cluster 20, as well as cluster 6 (***Figure 7—figure supplement 1a***) suggesting that it may be a common marker for the entire MB, in addition to the PMd. Finally, we examined expression of other markers (*Pitx2, Lhx1, Lhx5, Cdh11, Sim1, Sim2,* and *Nkx2.1*) that have previously been implicated in MB development (***Kim et al., 2020***; ***Kimura et al., 1996***; ***Bedont et al., 2015***; ***Puelles et al., 2000***; ***Martin et al., 2002***; ***Marion et al., 2005***; ***Shimogori et al., 2010***; ***Skidmore et al., 2012***; ***Miquelajáuregui et al., 2015***; ***Szabó et al., 2015***; ***Ferran et al., 2015***) and found significant alignment with clusters 1–5 for virtually all markers (***Figure 7—figure supplement 1a***).

## Spatial segregation of transcriptionally-distinct medial mammillary subpopulations

We further investigated differentially expressed genes among VPH^GLUT^ clusters 1–5 (***Figure 8a***), that comprise the medial mammillary region in an effort to determine if they exhibit any spatial organization with respect to the cytoarchitecurally-defined MnM, MM, and ML subdivisions. We identified a

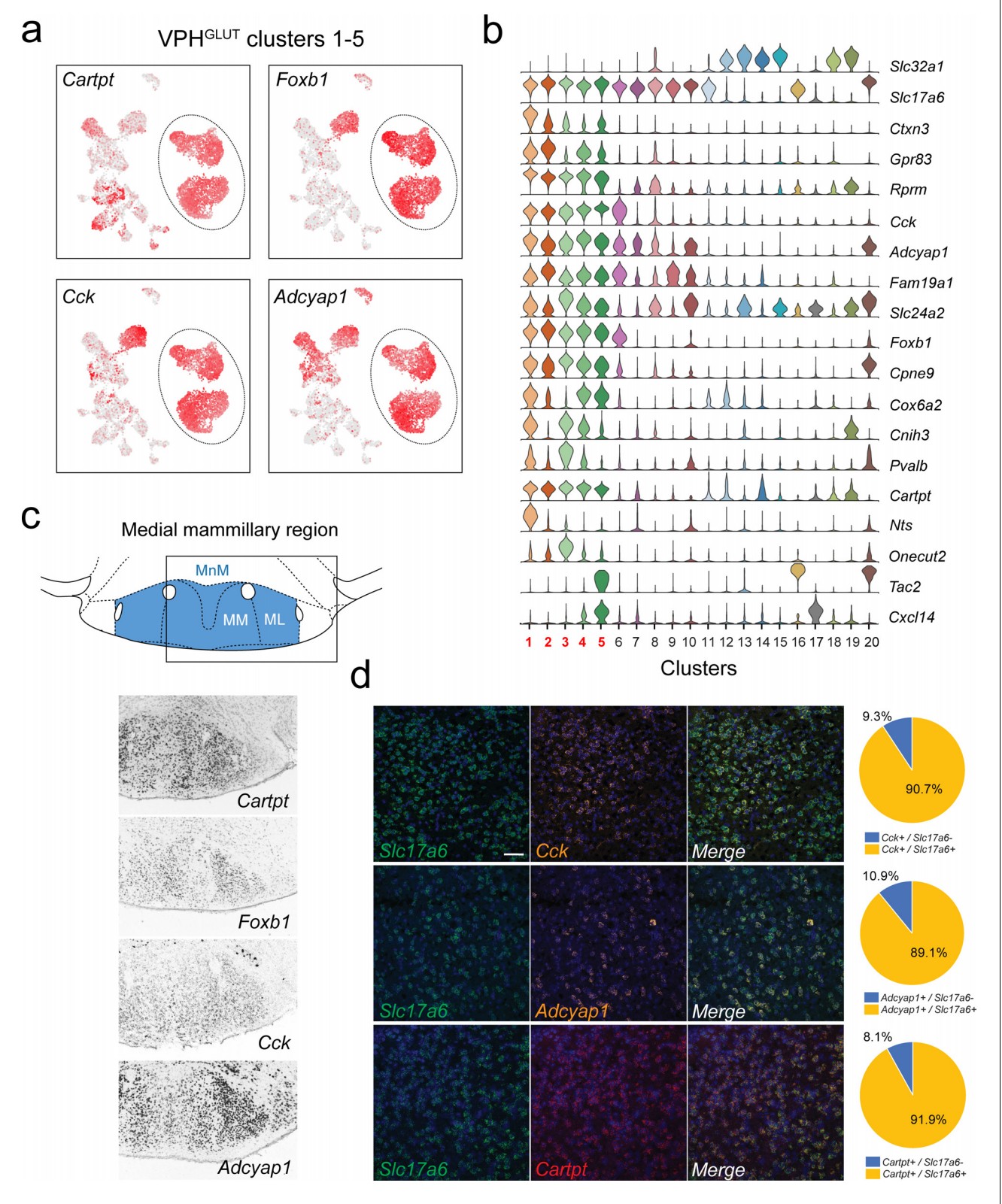

**Figure 7.** Identification of global markers for neurons in the medial mammillary region. (**a**) UMAP plots showing normalized expression of *Tac2, Tcf4, Cplx1,* and *Pvalb* enriched in VPH^GLUT clusters 1–5 (circled). (**b**) Violin plot showing discriminatory marker genes enriched in VPH^GLUT clusters 1–5 following *Slc32a1* and *Slc17a6* (top). (**c**) Mouse brain atlas schematic, modified from *Paxinos, 2012*, showing the medial mammillary, and its anatomical subdivisions (MnM, MM, and ML) in a coronal section at distance from bregma of −2.92 mm (top). ISH images for *Cartpt, Foxb1, Cck,* and *Adcyap1*

*Figure 7 continued on next page*

*Figure 7 continued*

from the ABA (*Lein et al., 2007*; bottom) showing widespread expression throughout the medial mammillary. (**d**) Confocal micrographs (40×) of FISH in coronal sections of wild type mice and corresponding pie charts representing co-expression of mRNA for *Slc17a6* and *Cck* (n = 2787 cells, three mice; upper), *Slc17a6* and *Adcyap1* (n = 1805 cells, three mice; middle) and *Slc17a6* and *Cartpt* (n = 2582 cells, three mice; lower). Scale bar (applicable to all micrographs) 50 μm.

The online version of this article includes the following figure supplement(s) for figure 7:

**Figure supplement 1.** Other key discriminatory markers enriched in the MB.

suite of differentially-expressed genes that define each of the transcriptionally distinct medial mammillary clusters 1–5 (*Figure 8b*) with two discriminatory genes from each shown in UMAP plots (*Figure 8c*). We went on to cross reference these makers with their spatial distribution within the medial mammillary region using the ABA (*Lein et al., 2007*) and found striking patterns of segmentation. Markers for cluster 1 neurons (*Nts*+, *Col25a1*+) appear to be enriched in the MM subdivision, medial to the principle mammillary tract (pm), whereas markers for cluster 4 neurons (*Nos1*+, *Calb1*+) are found concentrated in the ML subdivision, lateral to the principle mammillary tract (pm) and medial to the fornix (f). Cluster 2 neurons (*Gpr83*+, *Spock3*+) are concentrated in the central portion of the MnM subdivision, whereas cluster 3 neurons (*Pvalb*+, *Slc24a2*+) are enriched in the more dorsal portion of the MnM. Interestingly, cluster 5 neurons (*Tac2*+, *Cxcl14*+) appear to correspond to a thin rim of small diameter, *Tac2*+ neurons hugging the basal surface of the MM and ML (*Figure 8d*), overlapping with the region typically referred to as the TMNv or VTM (*Paxinos, 2012*). These *Tac2*+ neurons may be distinguished from the larger diameter *Tac2*+ neurons confined to the LM (VPH$^{GLUT}$ cluster 20). Despite being in the vicinity of TMNv HA neurons, 89.3% (176/197) of cluster 5 neurons are *Tac2*+ whereas only 2% (4/197) are *Hdc*+ indicating little to no overlap with HA neurons.

To further explore cellular heterogeneity among medial mammillary neurons, we subjected clusters 1–5 to another iteration of clustering which resulted in 12 subclusters (*Figure 8—figure supplement 1a*), each defined by suites of discriminatory markers (*Figure 8—figure supplement 1b,c*). Each of clusters 1–4 parsed into two to three subclusters, while cluster 5 remained largely intact. This further subclustering revealed markers that are common to single clusters and the discriminatory genes that define subclusters. For example, cluster 1 MM neurons are composed of three subclusters (3, 4, and 5) for which *Nts* and *Col25a1* are common to all, as shown previously (*Figure 8c, d*), while *Calb2* uniquely defines one subcluster, and *Adra2a* is enriched in another (*Figure 8—figure supplement 1c,d*). In sum, we found that richly diverse, transcriptionally distinct neuronal populations that comprise the medial mammillary region also exhibit a remarkable degree of compartmentalization within the known anatomical subdivisions of the region, demarcated by the principle mammillary tract and fornix.

## Topographically-distinct projection targets of genetically-defined MM and ML neurons in the thalamus and midbrain

We next asked whether the cluster-specific discriminatory genes that we identified in our scRNA-seq analysis of the MB (summarized in *Figure 9a*) may be used to genetically target specific anatomical subdivisions of the MB (summarized in *Figure 9b*) and shed light on the organization of their projection targets. Neurons in the MB make two highly specific projections in the brain: (1) a major, unidirectional projection to the anterior thalamic nuclei (ATN) through the mammillothalamic tract (mtt); and (2) a bidirectional connection to the ventral tegmental nucleus of Gudden (VTg), via the mamillotegmental tract (mtg) (summarized in *Figure 9c*; *Vann and Aggleton, 2004*; *Vann, 2010*; *Aggleton et al., 2010*; *Dillingham et al., 2015*; *Vann and Nelson, 2015*). The rodent ATN itself may be subdivided into three broad compartments, anteroventral (AV), anteromedial (AM), and anterodorsal (AD), each of which exhibit unique patterns of connectivity. In particular, the MM projects topographically to the AM, the ML to the AV and the LM to the AD (*Aggleton et al., 2010*; *Shibata, 1992*; *Jankowski et al., 2013*; *Bubb et al., 2017*; *Seki and Zyo, 1984*; *Hayakawa and Zyo, 1989*; *Wright et al., 2013*). We therefore set out to test our prediction that the transcriptionally-distinct cell populations that we found to be segregated within specific subcompartments of the MB, exhibit a similar topographic mapping to the ATN. To this end, we used three Cre recombinase mouse driver lines to target separate anatomical subregions of the MB by bilaterally injecting the

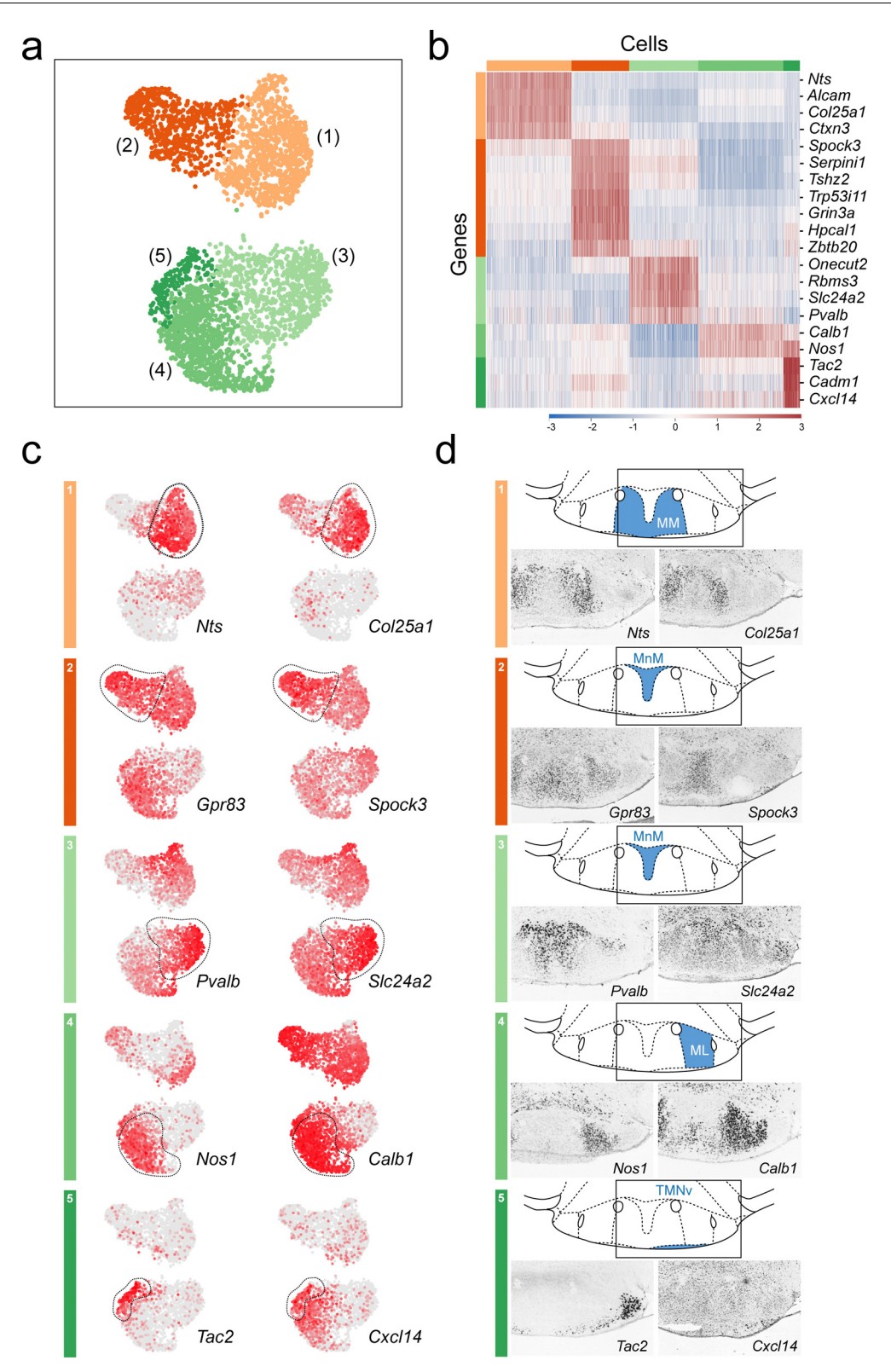

**Figure 8.** Anatomical compartmentalization of transcriptionally-distinct medial mammillary subpopulations. (a) UMAP plot showing just VPH[GLUT] clusters 1–5 (detail of *Figure 2b*). (b) Heatmap showing scaled expression of discriminatory genes across only VPH[GLUT] clusters 1–5. (c) Pairs of UMAP plots showing normalized expression of two discriminatory marker genes for each of VPH[GLUT] clusters 1–5 (top to bottom): cluster 1 (*Nts*, *Col25a1*), cluster 2 (*Gpr83*, *Spock3*), cluster 3 (*Pvalb*, *Slc24a2*), cluster 4 (*Nos1*, *Calb1*), and cluster 5 (*Tac2*, *Cxcl14*). (d) Mouse brain atlas schematics, modified from

*Figure 8 continued on next page*

*Figure 8 continued*

*Paxinos, 2012*, showing subregions of the medial mammillary in a coronal section at distance from bregma of −2.92 mm (top). ISH images for each of the two markers shown in (c) from the ABA (*Lein et al., 2007*; bottom).

The online version of this article includes the following figure supplement(s) for figure 8:

**Figure supplement 1.** Subclustering of MM neurons (VPH$^{GLUT}$ clusters 1–5).

MB with the viral anterograde tracer AAV-DIO-ChR2-EYFP (schematic in *Figure 9d*). Using the *Slc17a6* (VGLUT2)-Cre driver line to broadly target VPH$^{GLUT}$ neurons in the MB and their projections, we found labeling throughout the medial mammillary region at the injection site (*Figure 9e*), but largely sparing the LM, shown in a representative mouse. In addition to finding a high density of labeled axons in the mtt and mtg, we observed a plexus of fibers in the VTg and a high density of fibers in much of the ATN (AV and AM, but not AD, consistent with the latter being the target of LM projections). Next, using the *Nts*-Cre line to selectively target VPH$^{GLUT}$ cluster 1 neurons, that we had previously mapped to the MM subdivision, we found labeling at the injection site between the MM and ML and labeled axons in the mtt, mtg, and VTg, shown in a representative mouse. However, in the ATN, the AM subdivision appeared to be selectively targeted, while sparing the AV and AD (*Figure 9e*). Finally, using the *Calb1*-Cre driver to target VPH$^{GLUT}$ cluster 4 neurons, that we had mapped to the ML subdivision, we found clear labeling in the ML at the injection site (with some weak labeling in the MM/MnM) and labeled axons in the mtt, mtg, and VTg, shown in a representative mouse. Interestingly, and in stark contrast to our *Nts*-cre results, we observed highly specific labeling of the AV subdivision of the ATN, while sparing the AM and AD. These data are consistent with immunohistochemical evidence in guinea pig suggesting that CART (encoded by *Cartpt*) and calbindin (encoded by *Calb1*) are co-localized in the ML region of the MB and calbindin-immunoreactive fibers are concentrated in the AV subdivision of the ATN (*Żakowski et al., 2014*). Taken together, these anterograde tracing data suggest that not only are molecularly-defined subpopulations of MB neurons highly segregated within the anatomical subdivisions of the MB (MnM, MM, ML, and LM), but these subpopulations appear to project to distinct domains within the ATN, consistent with previous anterograde and retrograde tracing data without genetic specificity (*Aggleton et al., 2010*; *Shibata, 1992*; *Jankowski et al., 2013*; *Bubb et al., 2017*; *Seki and Zyo, 1984*; *Hayakawa and Zyo, 1989*; *Wright et al., 2013*).

## Discussion

In this work, we performed a systematic single-cell transcriptomic census of the molecular and spatial organization of cell types in the murine VPH. Using droplet-based scRNA-seq of >16,000 cells, we identified the molecular markers that define 20 neuronal and 18 non-neuronal cell clusters. We identified a diversity of mostly novel, and some previously identified, neuronal cell populations, many of which could be mapped to clearly defined anatomical compartments within the VPH (including the Arc, TMN, SUM, PMd, PMv, and approximately six subdivisions of the MB including the MnM, MM, ML, and LM; summarized in *Figure 10a–c*), using both ISH data from the ABA and multiplexed FISH. In particular, we observed that transcriptionally-distinct MB neurons are both confined to anatomically segregated compartments and, using three genetically-targeted neuronal populations in the MB, appear to project to precise anatomical targets in the ATN. This molecular census provides a rich resource for interrogating genetically-defined VPH circuits and their myriad roles in physiology and behavior.

In our unsupervised analysis of VPH neuronal populations, numerous molecularly distinct clusters emerged, but a broader dichotomy could be found based on the expression of genes necessary for the synthesis and vesicular release of GABA and glutamate (*Figure 2a*). The primary discriminatory genes in this regard are the largely mutually exclusive expression of *Slc32a1* (VGAT) and *Slc17a6* (VGLUT2) resulting in our identification of 13 populations of nominally glutamatergic neurons and 6 populations of nominally GABAergic neurons. Overall, the broad division between nominally GABAergic and glutamatergic populations aligns with observations from other recent scRNA-seq analyses of whole mouse brain (*Zeisel et al., 2018*; *Saunders et al., 2018*), whole hypothalamus (*Chen et al., 2017*; *Kim et al., 2020*) and hypothalamic subregions (*Mickelsen et al., 2019*;

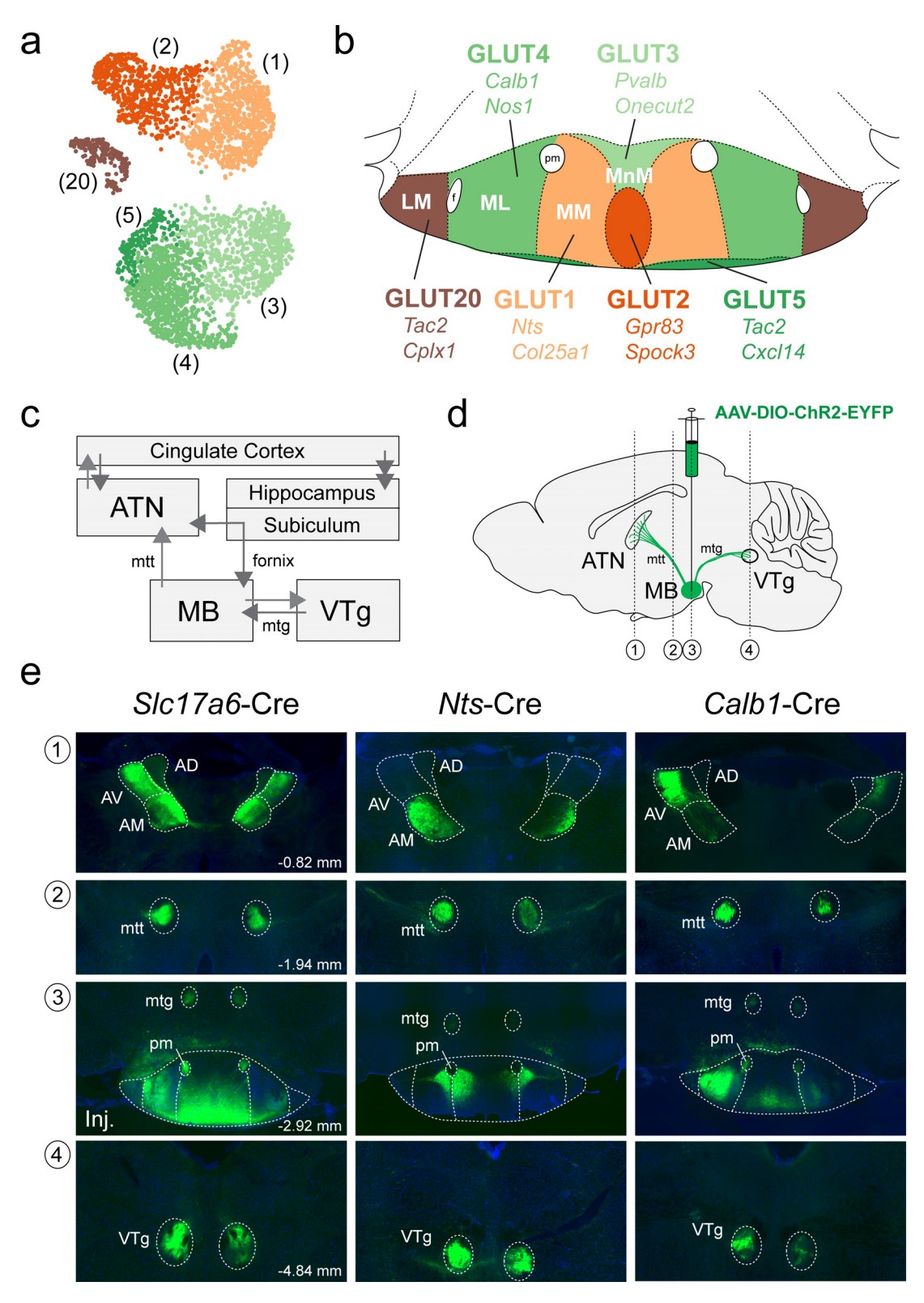

**Figure 9.** Anterograde projections of transcriptionally-distinct medial mammillary subpopulations to the anterior thalamus and midbrain. (a) UMAP plot showing just VPH[GLUT] clusters 1–5 and 20 (details from *Figure 2b*). (b) Mouse brain atlas schematic modified from *Paxinos, 2012*, showing subregions of the MB in a coronal section at distance from bregma of −2.92 mm. Color-coding of subdivisions indicate correlations between transcriptionally distinct subpopulations in (a) and anatomical subdivisions within the MB: cluster 1 (MM, light orange), cluster 2 (ventromedial MnM, dark orange),

*Figure 9 continued on next page*

*Figure 9 continued*

cluster 3 (dorsomedial MnM, light green), cluster 4 (ML, medium green), cluster 5 (TMNv or ventrobasal MM/ML, dark green) and cluster 20 (LM, brown). (c) Schematic of connections between the MB, hippocampal formation, ATN and VTg comprising the Papez circuit. (d) Schematic parasagittal section illustrating anterograde viral tracing using AAV-DIO-ChR2-EYFP injection in the MB of three cre driver lines (*Slc17a6*-Cre, *Nts*-Cre, and *Calb1*-Cre). Number labels indicate imaged coronal sections. (e) Representative fluorescence images of coronal sections for anterograde tracing in *Slc17a6*-Cre (n = 3 mice; left), *Nts*-Cre (n = 3 mice; middle) and *Calb1*-Cre (n = 2; right) mice at the following approximate distances from bregma: (1) ATN, −0.82 mm; (2) mtt, −1.94 mm; (3) MB injection site, −2.92 mm; and (4) VTg, −4.84 mm. All sections were counterstained with DAPI (blue). Abbreviations: AV (anteroventral), AM (anteromedial), AD (anterodorsal), mtt (mammilothalamic tract), mtg (mammillotegmental tract), and pm (principle mammillary tract).

*Moffitt et al., 2018*; *Romanov et al., 2017*; *Campbell et al., 2017*; *Rossi et al., 2019*; *Kim et al., 2019*). One notable exception is VPH$^{HA}$ cluster 17, which expresses low levels of both *Slc32a1* and *Slc17a6*, and we identified as having a histaminergic phenotype (*Figures 2* and *5* and *Figure 5—figure supplement 1*). Another notable exception is a putative dual identity cluster 8 subcluster, which exhibits high co-expression of both *Slc32a1* and *Slc17a6*, and corresponds well to markers of the SUM (*Figure 4—figure supplement 1*) and, possibly, to a known dual phenotype, hippocampal-projecting population previously identified in the SUM (*Pedersen et al., 2017*; *Boulland et al., 2009*; *Soussi et al., 2010*; *Hashimotodani et al., 2018*).

Within each fast neurotransmitter identity, discrete classifications of VPH neurons were arrived at based on a suite of other discriminatory markers with few classifications being made on the basis of single markers alone. Most neuronal clusters are also defined by a combination of neuropeptide transcripts, which are associated with both well-known and novel populations. The co-expression of both fast neurotransmitter and multiple neuropeptide markers in single neurons is suggestive of a capacity for co-transmission (*van den Pol, 2012*). Other important cluster-specific markers include some combination of transcripts that encode transcription factors, calcium binding proteins, synaptic and extracellular matrix proteins, receptors and other gene categories. That these combinations of markers specify the identity of distinct populations of VPH neurons likely reflects a convergence of their unique molecular specification during development, their neurochemistry and specific synaptic connectivity within functional circuits.

While several of the diverse neuronal clusters that emerged from our scRNA-seq analysis were readily identifiable based on well-known signatures (*e.g. Kiss1+/Tac2+/Pdyn+* KNDy Arc neurons, *Agrp+/Npy+* Arc neurons, *Hdc+* TMN HA neurons, etc.), most were novel and differentially expressed genes required both cross-referencing with the ABA (*Lein et al., 2007*) and extensive multiplexed FISH to effectively trace them to anatomical regions within the VPH. We found that a number of novel neuronal cell populations exhibited striking compartmentalization within distinct VPH nuclei (*Figure 10b*). This was especially apparent in our mapping of VPH$^{GLUT}$ clusters 1–5 and 20 to subcompartments within the MB. The MB is a conserved diencephalic structure across rodents, primate models and humans and represents an important link between subicular outputs of the hippocampal formation (via the fornix) and the ANT (via the mtt), which itself has reciprocal projections with the cingulate/retrosplenial cortices. MB neurons also make bidirectional connections with the midbrain VTg (through the mtg; *Vann and Aggleton, 2004*; *Vann, 2010*; *Aggleton et al., 2010*; *Dillingham et al., 2015*; *Vann and Nelson, 2015*; *Jankowski et al., 2013*; *Bubb et al., 2017*). MB neurons, as a key component of this limbic pathway, are critical for spatial memory as lesions to the MB and its outputs through the mtt and mtg, result in severe deficits in spatial memory in animal models and human clinical data (*Vann and Aggleton, 2004*; *Vann, 2010*; *Dillingham et al., 2015*; *Vann and Nelson, 2015*; *Bubb et al., 2017*; *Vann and Aggleton, 2003*; *Vann, 2013*; *Tsivilis et al., 2008*). For example, Korsakoff's syndrome, caused by thiamine deficiency associated with chronic alcoholism, is characterized by selective degeneration of the MB and is accompanied by severe amnesia (*Kopelman, 1995*; *Kril and Harper, 2012*).

Consistent with our identification of VPH$^{GLUT}$ clusters 1–5 as representing the medial mammillary region, several of the discriminatory marker genes that emerged from our analysis result in spatial memory deficits when mutated. For example, *Foxb1* is well-known to define the MB during early embryonic development (*Shimogori et al., 2010*; *Kaestner et al., 1996*; *Wehr et al., 1997*; *Radyushkin et al., 2005*) and *Foxb1*-null mice exhibit dysgenesis of both the MB and mtt (*Wehr et al., 1997*; *Radyushkin et al., 2005*; *Alvarez-Bolado et al., 2000*) and display defects in

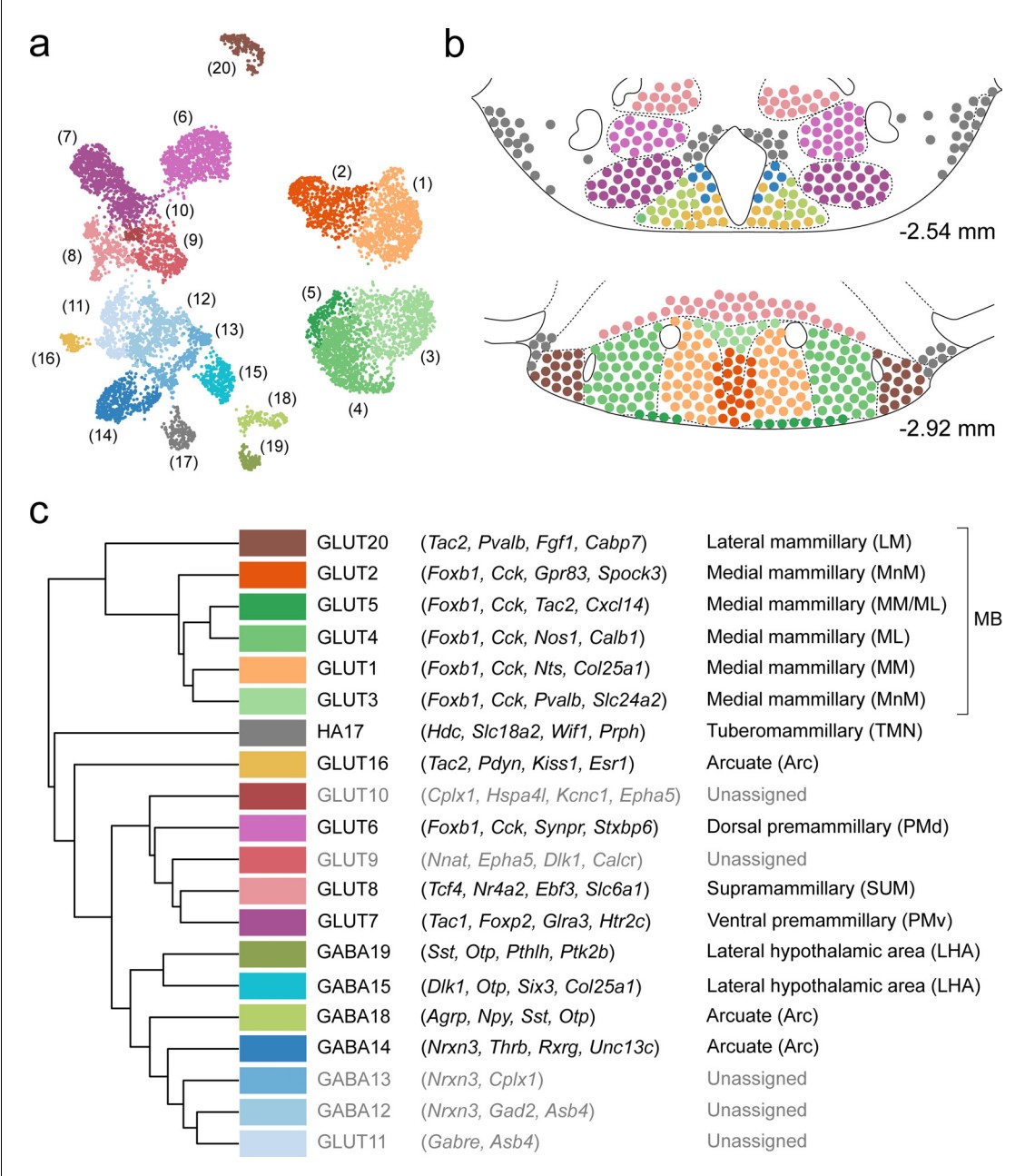

**Figure 10.** Summary of the relationship between transcriptionally distinct VPH cell clusters and anatomical parcellation of the VPH. (a) UMAP of all VPH neuronal clusters. (b) Mouse brain atlas schematic (*Paxinos, 2012*) showing two representative VPH coronal sections at distances from bregma of −2.54 and −2.92 mm. Neuronal cell types identified in scRNA-seq analysis that can be assigned to specific anatomical subregions of the VPH are indicated. (c) Hierarchical clustering of single-cell clusters (left), discriminatory markers (middle), and the anatomical region they correspond to (right) are listed.

spatial memory tasks, particularly spatial navigation and working memory (*Radyushkin et al., 2005*). These data suggest that the MB may have specific roles to play in spatial memory processing. Furthermore, the enrichment of the recently deorphanized receptor *Gpr83* (*Gomes et al., 2016*) that we observed in this region (*Figure 7* and *Figure 7—figure supplement 1*) aligns with previous anatomical work showing high expression in the MB (*Pesini et al., 1998*) and behavioral experiments showing that *Gpr83*-null mice also exhibit impaired spatial learning (*Vollmer et al., 2013*). These data indicate that VPH[GLUT] clusters 1–5 share a suite of common markers that align well with the known development and function of the medial mammillary region of the MB.

Interestingly, a selection of the defining marker genes for the medial mammillary region (clusters 1–5; *Figures 7* and *8*, *Figure 7—figure supplement 1* and *Figure 8—figure supplement 1*) was also found to be enriched in some combination of other clusters we ascribed to the LM (cluster 20), PMd (cluster 6), PMv (cluster 7), SUM (cluster 8), and others. For example, we found that while *Cck*, *Foxb1*, *Pitx2*, *Lhx1*, *Lhx5*, *Sim1* are indeed all enriched in clusters 1–5, they are all additionally expressed in cluster 6, among others. In addition to the MB, *Foxb1* is known to also be expressed in the developing PMd (*Wehr et al., 1997*; *Alvarez-Bolado et al., 2000*), which is linked anatomically and functionally with the MB, including a parallel projection to the ATN (*Canteras et al., 2008*; *Alvarez-Bolado et al., 2000*; *Canteras and Swanson, 1992*). Furthermore, *Foxb1*-null mice not only exhibit defects in spatial memory (*Radyushkin et al., 2005*) but abnormal nurturing behavior including pup retrieval and nest-building (*Wehr et al., 1997*). These data suggest a possible defect in PMv-associated reproductive behavior (*Donato and Elias, 2011*) and/or PMd-associated threat-response or defensive behavior (*Canteras et al., 2008*; *Cezario et al., 2008*; *Blanchard et al., 2003*). Taken together, our transcriptomic data reinforce the notion that the MB, PMd, PMv and perhaps other surrounding VPH structures, share an anatomical and functional interrelationship, rooted in their shared developmental history (*Ferran et al., 2015*; *Bedont et al., 2015*).

In mapping differentially expressed markers of VPH$^{GLUT}$ clusters 1–5 and 20 to the MB, we found that the MB is not only more transcriptionally diverse than previously appreciated, but distinct MB neurons also exhibit a remarkable degree of segregation within long-established, anatomically-defined subdivisions of the MB. While the primary anatomical division in the MB is between medial and lateral, the medial mammillary may be further subdivided into multiple subcompartments (MnM, MM, and ML) on the basis of Nissl and Golgi staining, cell morphology, white matter boundaries (pm, fornix, etc.) and patterns of connectivity (*Paxinos, 2012*; *Jankowski et al., 2013*; *Bubb et al., 2017*; *Seki and Zyo, 1984*; *Allen and Hopkins, 1988*; *Allen and Hopkins, 1989*). However, the molecular basis for cell type diversity and the spatial organization of cell types within the MB is poorly understood. Although a recent scRNA-seq analysis of the whole hypothalamus captured the VPH, it resolved only a single population of *Foxb1+/Cartpt+* neurons assigned to the MB (*Chen et al., 2017*). Most recently, a much larger scale analysis of the developing mouse hypothalamus identified several MB populations (*Kim et al., 2020*). We resolved at least six distinct populations (a total of 15 subclusters following a second iteration of clustering) that we subsequently mapped to six anatomical subdivisions of the MB (summarized in *Figures 9a,b* and *10*). For example, VPH$^{GLUT}$ cluster 20 (*Tac2+/Cplx1+*) is confined to the LM and is distinct from cluster 4 (*Nos1+/Calb1+*), which is confined to the ML, which itself is distinct from cluster 1 (*Nts+/Col25a1+*), which broadly corresponds to the MM. The patterns of discriminatory genes for clusters 2 (*Gpr83+/Spock3+*) and cluster 3 (*Pvalb+/Slc24a2+*) are more complex. Cluster 3 markers appear to correspond to the more dorsal MnM, whereas cluster 2 markers are more ventromedially located, bounded by the MM on either side, hence the slightly modified boundaries schematized in *Figures 9b* and *10b*. Cluster 5 (*Tac2+/Cxcl14+*) appears to correspond to a thin rim of neurons on the ventrobasal surface of the MB and distinct from *Tac2+* neurons in the LM. The high degree of regional segmentation is striking, especially juxtaposed with the highly diffuse spatial organization of transcriptionally-distinct neuronal populations in the adjacent LHA (*Mickelsen et al., 2019*).

The significance of this highly compartmentalized pattern of gene expression may lie in the specific connectivity between MB subcompartments and other regions of the brain. The importance of MB projections to the ATN and VTg is highlighted by previous work showing that specific lesions to the mtt and VTg, but not to the postcommissural fornix, result in deficits in spatial memory, suggesting that MB-ATN and MB-VTg connectivity have critical roles in memory, likely independent of hippocampal input (*Vann, 2013*). We found that genetically-defined MB neurons, using markers we identified in our scRNA-seq analysis to specify MB subpopulations, project to distinct thalamic and midbrain targets. In particular, while *Slc17a6+* MB neurons project to the AV and AM, *Nts+* MM neurons project specifically to the AM, while *Calb1+* ML neurons project specifically to the AV. All three MB projections target the VTg while none targeted the AD. Our genetically-targeted anterograde tracing is consistent with the known topography of MB-ATN projections (*Aggleton et al., 2010*; *Shibata, 1992*; *Jankowski et al., 2013*; *Bubb et al., 2017*; *Seki and Zyo, 1984*; *Hayakawa and Zyo, 1989*; *Wright et al., 2013*) and suggests that the spatial architecture of these circuits may be explained by an underlying molecular organization. Furthermore, the AM, AV, and AD subregions of the ATN each have differential patterns of reciprocal connectivity with different cortical regions and

subcortical regions and are thought to comprise parallel sub-circuits within a brain-wide memory network (*Aggleton et al., 2010*; *Jankowski et al., 2013*). For example, aside from reciprocal connections with the anterior cingulate, the AM is uniquely interconnected with the prelimbic cortex, and thought to have a role in relaying information from the hippocampal formation and MB to prefrontal cortical centers associated with higher cognitive function (*Aggleton et al., 2010*; *Jankowski et al., 2013*). The AV, in contrast, is interconnected with the retrosplenial cortex (*Shibata and Kato, 1993*) and subiculum (*Shibata, 1993*), and implicated in theta entrainment (*Aggleton et al., 2010*; *Jankowski et al., 2013*). Aside from topographic ATN projections, the spatial organization of MB circuits is also likely a reflection of topographic projections from specific subicular subregions/cell types (*Allen and Hopkins, 1989*; *Christiansen et al., 2016*; *Bienkowski et al., 2018*), which also exhibit a broader molecular and spatial organization (*Cembrowski et al., 2018*). Future work combining transcriptomic profiling, functional connectomics, manipulation and monitoring of genetically-defined neurons in the subiculum, MB, VTg, ATN, and cingulate/retrosplenial cortices may reveal an overarching organization of parallel circuits within this brain-wide memory system. Recent large-scale efforts towards molecular profiling and detailed connectivity analysis among other thalamic pathways, have revealed novel insights into the overarching logic of thalamic organization (*Harris et al., 2019*; *Phillips et al., 2019*). Taken together, our transcriptomic analysis of MB subpopulations may shed light on the construction of MB circuits during development, their differential connectivity, regulation of excitability, repertoire of synaptic signaling mechanisms and specific functional roles in spatial memory.

Finally, given the important and conserved role of the MB as a node in a brain-wide memory system and multiple aspects of spatial and episodic memory, it is significant that the MB may have a key role in the pathogenesis of Alzheimer's Disease (AD). Evidence from imaging of AD patients and postmortem examination of AD-afflicted brains suggest that the MB, fornix, and mtt all show varying degrees of atrophy or degeneration that may be correlated with cognitive decline and memory loss (*Callen et al., 2001*; *Copenhaver et al., 2006*; *Grossi et al., 1989*; *Baloyannis et al., 2016*). Recent work has implicated the MB and other components of the Papez circuit as sites of particular vulnerability in the pathogenesis of AD in mice (*Canter et al., 2019*). Using a quantitative measure of beta-amyloid deposition in the whole brain of 5XFAD mice, a mouse model of AD (*Oakley et al., 2006*), the MB was identified as among the earliest sites of beta-amyloid deposition in the brain, appearing as early as two months after birth. This observation was accompanied by an increase in medial mammillary neuronal excitability, that when chemogenetically silenced, resulted in reduced beta amyloid deposition (*Canter et al., 2019*). In addition, other VPH subpopulations may exhibit selective vulnerability in AD as evidenced by the presence of AD pathological features in the TMN and dramatic loss of TMN HA neurons in *post mortem* AD-afflicted brains (*Saper and German, 1987*; *Airaksinen et al., 1991*; *Nakamura et al., 1993*; *Oh et al., 2019*), likely contributing to sleep-wake disturbances commonly observed in AD patients (*Spira et al., 2014*). Taken together, these clinical observations and intriguing evidence from disease models underscore the critical need to elucidate both the basic biology of the MB and other VPH cell types, the circuits they give rise to and their potential vulnerability in the early stages of AD pathogenesis.

Overall, our analysis of the molecular and spatial organization of VPH cell types provides the foundation for a more detailed understanding of the cellular composition and wiring diagram of the VPH. In particular, our analysis reveals the molecular underpinnings of the modular organization of both the MB and its topographic projections to subregions of the anterior thalamus as well as the midbrain. In addition, this work serves as a rich resource for cell type-specific deconstruction of VPH circuit function, through pharmacological tools and the use of genetically-targeted manipulation and monitoring methodology. Overall, our VPH cell type census, along with other recent hypothalamic scRNA-seq analyses (*Mickelsen et al., 2019*; *Moffitt et al., 2018*; *Chen et al., 2017*; *Zeisel et al., 2018*; *Romanov et al., 2017*; *Campbell et al., 2017*; *Rossi et al., 2019*; *Kim et al., 2019*), contributes to a more comprehensive picture of the cell type diversity and spatial organization of the mammalian hypothalamus, as well as informing the molecular, cellular, and synaptic mechanisms of hypothalamic circuit function in health and disease.

# Materials and methods

**Key resources table**

| Reagent type (species) or resource | Designation | Source or reference | Identifiers | Additional information |
|---|---|---|---|---|
| Strain (*Mus musculus*) | C57Bl/6 | The Jackson Laboratory | Cat# JAX:000664 RRID:IMSR_JAX:000664 | - |
| Genetic reagent (*Mus musculus*) | *Slc17a6*-Cre | The Jackson Laboratory | Cat# JAX:016963 RRID:IMSR_JAX:016963 | - |
| Genetic reagent (*Mus musculus*) | *Nts*-Cre | The Jackson Laboratory | Cat#: JAX:017525 RRID:IMSR_JAX:017525 | - |
| Genetic reagent (*Mus musculus*) | *Calb1*-Cre | The Jackson Laboratory | Cat#: JAX:028532 RRID:IMSR_JAX:028532 | - |
| Recombinant DNA reagent (Adeno - associated Virus) | AAV2-Ef1α-DIO-hChR2(H134R)-EYFP | UNC Viral Core | RRID:Addgene_55640 | Diesseroth Lab |
| Peptide, recombinant protein | Protease XXIII | Sigma | Cat# P4032 | 2.5 mg/mL |
| Peptide, recombinant protein | Trypsin inhibitor | Sigma | Cat# T9253 | 10 mg/mL |
| Commercial assay or kit | Chromium Single Cell 3' Library and Gel Bead Kit | 10x Genomics | Cat# 1000075 | V2 and V3 |
| Commercial assay or kit | RNAScope fluorescent multiplex detection assay reagents | ACDBio | Cat# 320851 | V1 |
| Commercial assay or kit | RNAScope V1 fluorescent multiplex detection assay probes | ACDBio | Cat# 405911 Cat# 313641 Cat# 432001 Cat# 402271 Cat# 482531 Cat# 490471 Cat# 450291 Cat# 421931 Cat# 319171 Cat# 410351 Cat# 446391 Cat# 537191 Cat# 412361 | *Adcyap1* *Calb2* *Cartpt* *Cck* *Cplx1* *Hdc* *Maob* *Pvalb* *Slc17a6* *Tac1* *Tac2* *Tcf4* *Wif1* |
| Software, algorithm | Fiji | PMID:22743772 | RRID:SCR_002285 | https://imagej.net/Fiji |
| Software, algorithm | Photoshop | Adobe | RRID:SCR_014199 | - |
| Software, algorithm | Illustrator | Adobe | RRID:SCR_010279 | - |
| Software, algorithm | Scanpy | Python | RRID:SCR_018139 | v1.3.7 |
| Software, algorithm | Cell Ranger | 10x Genomics | RRID:SCR_017344 | v3.0.2 |
| Other | Vectashield Hardset Mounting Media w/DAPi | Vector Labs | Cat# H-1500 | - |
| Other | ProLong Gold Mounting Media w/DAPi | Thermofisher | Cat# P36965 | - |

## Animals

To collect VPH neurons scRNA-seq analysis as well as fluorescence in situ hybridization experiments, we used both male and female C57BL/6 (JAX stock #000664) mice. For anterograde tracing experiments, we used the following Cre recombinase driver lines: (1) *Slc17a6*tm2(cre)Lowl/J knock-in mutant mice (JAX stock #016963, referred to here as *Slc17a6*-Cre mice; *Vong et al., 2011*). (2) *Nts*-Cre knock-in mutant mice (B6;129-Nts tm1(cre)Mgmj/J; JAX stock # 017525; *Leinninger et al., 2011*). (3) *Calb1*-IRES2-Cre-D knock-in mutant mice (B6;129S-Calb1 tm2.1(cre)Hze/J; JAX stock # 028532, referred to here as *Calb1*-Cre). All mice were fed ad libitum and kept on a 12 hr light-dark cycle.

## scRNA-seq cell capture and sequencing

Hypothalamic brain slices were obtained from juvenile (p30-34) mice, from a total of five male mice and five female mice over two independent harvests. The first harvest consisted of three male (pooled), three female (pooled), while the second consisted of two males (pooled) and two females (pooled), following previously described procedures (*Mickelsen et al., 2019*). Briefly, mice were anesthetized with isoflurane, then rapidly sacrificed by decapitation during the same time period (morning, 09:00-11:00). Brain slices were cut using a vibrating microtome (Lafayette Instrument Company) at a thickness of 225 μm in ice-cold high-sucrose slicing solution consisting of the following components (in mM): 87 NaCl, 75 sucrose, 25 glucose, 25 NaHCO$_3$, 1.25 NaH$_2$PO$_4$, 2.5 KCl, 7.5 MgCl$_2$, 0.5 CaCl$_2$ and 5 ascorbic acid saturated with 95% O$_2$/5% CO$_2$. Slices were then enzyme-treated for ~15 min at 34°C with protease XXIII (2.5 mg/mL; Sigma) in a high-sucrose dissociation solution containing the following (in mM): 185 sucrose, 10 glucose, 30 Na$_2$SO$_4$, 2 K$_2$SO$_4$, 10 HEPES buffer, 0.5 CaCl$_2$, 6 MgCl$_2$, 5 ascorbic acid (pH 7.4) and 320 mOsm. Slices were washed three times with cold dissociation solution then transferred to a trypsin inhibitor/bovine serum albumin (TI/BSA) solution (10 mg/mL; Sigma) in cold dissociation solution. Four to five slices were obtained from each animal that approximately corresponded to mouse brain atlas images representing the distance from bregma −2.54,–2.70, −2.92, and −3.16 mm which includes the VPH. The VPH was dissected from each slice using a fine scalpel and forceps under a dissecting microscope and each slice was imaged and subsequently mapped onto mouse atlas images (*Figure 1b*; *Paxinos, 2012*). Microdissected tissue was kept in cold TI/BSA dissociation solution until trituration. Immediately before dissociation tissue was incubated for ~10 min at 37°C, then triturated with a series of small bore fire-polished Pasteur pipettes. Single-cell suspensions were passed through a 40 μm nylon mesh filter to remove any large debris or cell aggregates and kept on ice until single-cell capture.

Cell viability for each sample was assessed on a Countess II automated cell counter (Thermo-Fisher), and up to 12,000 cells were loaded for capture onto an individual lane of a Chromium Controller (10x Genomics). Single-cell capture, barcoding and library preparation were performed using the 10x Chromium platform (*Zheng et al., 2017*) according to the manufacturer's protocol (#CG00052) using version 2 (V2) chemistry for the first set of male and female samples and version 3 (V3) chemistry for the second set. cDNA and libraries were checked for quality on Agilent 4200 Tapestation, quantified by KAPA qPCR. The two V2 chemistry libraries were sequenced on individual lanes of an Illumina HiSeq4000 and the V3 chemistry libraries were pooled at 16.67% of lane of an Illumina NovaSeq 6000 S2 flow cell each, all samples targeting 6,000 barcoded cells with an average sequencing depth of 50,000 reads per cell.

## scRNA-seq data processing, quality control, and analysis

Illumina base call files for all libraries were converted to FASTQs using bcl2fastq v2.20.0.422 (Illumina) and FASTQ files were aligned to the mm10 (GRCh38.93, 10× Genomics mm10 reference 3.0.0) using the version 3.0.2 Cell Ranger count pipeline (10x Genomics), resulting in four gene-by-cell digital count matrices. Downstream analysis was performed using Scanpy (v1.3.7; *Wolf et al., 2018*) and can be found in its entirely on GitHub (see Data and code availability). Initial quality control was performed on each library individually and cells were excluded from downstream analysis by the following criteria: fewer than 2,000 transcripts, fewer than 1,000 genes, more than 50 hemoglobin transcripts, and more than 15% mtRNA content. Genes with fewer than three counts in at least three cells were also excluded from downstream analysis. These filtering criteria resulted in a substantial increase in data set quality but a dramatic reduction in called cells; the resulting individual

counts matrices were reduced to 5,191 and 3,223 cells for the male and female V2 libraries, respectively, and 4,622 and 3,955 cells for the male and female V3 libraries. These matrices were then concatenated together, resulting in an initial aggregated counts matrix of 16,991 cells by 20,202 genes. This aggregated counts matrix was normalized by the total number of counts per cell then multiplied by the median number of counts across all cells and finally $\log_2$ transformed.

The 1,500 most highly variable genes as measured by dispersion were selected for the computation of principal components (PCs). Prior to the computation of PCs, mitochondrial, ribosomal, hemoglobin, cell cycle (homologs of those defined in *Macosko et al., 2015*; *Campbell et al., 2017*) genes, and the specific genes *Xist*, *Fos*, *Fosb*, *Jun*, *Junb*, and *Jund* were excluded from this set of highly variable genes. The first 50 PCs were computed and subsequently corrected for batch effects related to 10× chemistry version and mouse sex using Harmony (*Korsunsky et al., 2019*; with theta_chemistry = 2, theta_sex = 0.5). Without this batch correction, clusters were stratified by 10x chemistry (shown in *Figure 1—figure supplement 1*). The 15 PCs with highest variance ratios were used to construct a k = 15 nearest-neighbor graph (k-NN graph, where distance is measured using cosine distance) and a 2D UMAP embedding was generated using this graph (min_dist = 0.5). Initial clusters were assigned via the Leiden community detection algorithm on this k-NN graph, resulting in 31 initial clusters, of which 20 remained after the removal of doublets (shown in *Figure 1—figure supplement 1f*).

Marker genes for each cluster, computed by a 'one-versus-rest' methodology comparing mean expression of every gene within a cluster to the expression in all other cells, were used to assign putative cell types to each cluster. For each cluster $c$ containing $n$ cells and each gene $g$ for which the mean expression $E_{cg} - E_{\neg cg} > 2$, we rank all cells by the expression of gene $g$. For a highly specific marker gene for cluster $c$, we expect the first $n$ ranks to correspond to cells of cluster $c$. We construct a ROC curve such that:

|  | Cell in cluster C | Cell not in cluster C |
|---|---|---|
| Rank <= $n$ | TP | FP |
| Rank > $n$ | FN | TN |

The area under this curve (*AUROC*) is computed for each cluster-gene ($c,g$) pair, and genes with *AUROC >0.8* are used to putatively identify cell types. Genes identified this way are consistent with genes identified using ScanPy (`scanpy.tl.rank_genes_groups` with the Wilcoxon Rank-Sum test).

Of the 31 initial clusters, several clearly exhibited signatures of two (or more) cell types and 713 cells from 11 clusters were excluded from further analysis. The remaining cells were classified as neuronal or non-neuronal using a simple two state Gaussian mixture model. Briefly, the median expression of *Snap25, Syp, Tubb3, and Elavl2* in each cluster was used to fit the model and classify the clusters as neuronal (high expression) and non-neuronal (low expression). We also tested classifying each individual cell using this model which lead to qualitatively equivalent classification. The cells for each classification were subsequently reanalyzed separately.

Neuronal cells underwent another round of filtering, with fewer than 3,500 transcripts and 2,200 genes being excluded. The raw expression matrix of the remaining 9,304 neuronal cells was normalized, batch corrected, and embedded with UMAP as described above with the only alteration being 20 PCs were used to build a k = 10 NN graph. Clustering with Leiden community detection led to 20 neuronal clusters. Non-neuronal cells were reanalyzed without additional filtering, and the expression matrix for the 6,069 non-neuronal cells was reanalyzed as described for the neuronal cells; Leiden clustering yielded 18 non-neuronal clusters.

In order to test the robustness of our neuronal clustering analysis and the specificity of the marker genes presented throughout the manuscript, we devised a computational experiment where we predict the cluster label of each cell using only the expression at a small set of neuronal marker genes. Briefly, we split our neuronal expression matrix and associated cluster labels into two groups, a training and a testing group, each with a proportional number of cells from each cluster. Using the first group, we recomputed the top two marker genes from each of the 20 neuronal clusters using the one-versus-rest methodology described above. The expression of the cells in the first group with this set of genes was used train a random forest classification model, a model that takes the

expression signature of a cell and predicts its cluster label. This model was then used to predict the cluster labels of the cells in the second group, which we compare to the original cluster labels. We built such a model using the top 2, 3, 5, 10, 15, and 20 genes from each cluster and predicted the cluster labels for the second group for using each model. This process is illustrated in *Figure 1—figure supplement 3a–g* and the source code is available in this project's GitHub repository (see Data and code availability).

### Fluorescence in situ hybridization (FISH)

FISH was carried out as previously described (*Mickelsen et al., 2019*). Briefly, to prepare tissue sections for FISH, male juvenile wild type C57BL/6 mice (P30–47) were anesthetized with isoflurane, decapitated and brains were dissected out into ice-cold sucrose. Brains were frozen on dry ice and cryosectioned at a thickness of 14 µm directly onto SuperFrost Plus microscope slides. Sections were then fixed with 4% paraformaldehyde (PFA) at 4°C for 15 min, and dehydrated for 5 min each in 50, 70, and 100% ethanol. RNAscope 2.5 Assay (Advanced Cell Diagnostics) was used for all FISH experiments according to the manufacturer's protocols (*Wang, 2012*). All RNAscope FISH probes were designed and validated by Advanced Cell Diagnostics. ISH images from the Allen Brain Institute were acquired from the publically available resource, the Allen Mouse Brain Atlas (www.mouse.brain-map.org/; *Lein et al., 2007*) and were acquired with minor contrast adjustments and converted to grayscale to maintain image consistency.

### Anterograde viral tract-tracing

For anterograde tracing experiments, male P30-42 *Slc17a6*-cre, *Nts*-Cre, and *Calb1*-Cre mice were bilaterally injected with 50–100 nL of AAV2-Ef1α-DIO-hChR2(H134R)-EYFP (UNC Viral Core, Diesseroth Lab) into the MB (anteroposterior (AP): −2.90 mm, mediolateral (ML): ±0.04 mm, dorsoventral (DV): −5.20 mm) and allowed to incubate for 4–6 weeks. *Slc17a6*-cre mice (n = 3) were injected with 100 nL while *Nts*-Cre mice (n = 3) and *Calb1*-cre mice (n = 2) were injected with 50 nL. For histology, mice were anesthetized with ketamine/xylazine and transcardially perfused with 10 mL of 0.125 M saline followed by 40 mL of 4% PFA in 1× PBS (pH 7.4). Brains were then dissected and post-fixed for 24 hr in 4% PFA, followed by cryoprotection in 30% sucrose for 48 hr. Brains were flash frozen with cold isopentane and stored at −80°C. Frozen brains were cut into 50 µm thick coronal sections containing the ATN, LHA, MB, and VTg on a cryostat (Leica 3050 s) into 1× PBS. Slices were then mounted onto slides with Vectashield Hardset mounting media containing DAPI (Vector Labs). Regions of interest were imaged at 10× magnification on a fluorescence microscope ( Keyence BZ-X700). Images were processed with ImageJ, Adobe Photoshop CS, and Adobe Illustrator CC. Based on *post hoc* histological evaluation of the injection site, mice were excluded from subsequent anatomical analysis if MB injection sites were absent or off-target.

### Data and code availability

Raw sequencing data and counts matrices are available in the Gene Expression Omnibus, accession number GSE146692. We further provide the final analyzed data sets in the form of H5AD (*Wolf et al., 2018*) and Loupe Cell Browser data files at https://singlecell.jax.org/hypothalamus. All code used to produce the analysis and figure panels is available on GitHub: https://github.com/The-JacksonLaboratory/ventral-posterior-hypothalamus-scrnaseq (*Flynn, 2020*; copy archived at https://github.com/elifesciences-publications/ventroposterior-hypothalamus-scrna-seq).

## Acknowledgements

We gratefully acknowledge R Kanadia, A Tzingounis, for helpful discussions and comments on the manuscript, and M Picciotto for helpful discussions. We also gratefully acknowledge D Luo for single-cell capture and scRNA-seq library preparation, A Mosley for genotyping support, C O'Connell for imaging support and M Samuels for sequencing support. We would also like to thank A Fujita, Y Huang, as well as all members of the Jackson Lab for support, assistance, and helpful discussions. We would also like to thank AL Lucido for additional support. Project supported by the National Institutes of Health grant R01MH112739 (to ACJ), a Connecticut Innovations Regenerative Medicine Research Fund grant 15-RMD-UCHC-01 (to PR), and an NIH Shared Instrumentation Grant S10OD016435 (to A Nishiyama) for imaging support.

## Additional information

### Competing interests

Mohan Bolisetty: Affiliated with Bristol-Myers Squibb. The author has no financial interests to declare. The other authors declare that no competing interests exist.

### Funding

| Funder | Grant reference number | Author |
| --- | --- | --- |
| National Institute of Mental Health | R01MH112739 | Alexander C Jackson |
| Connecticut Innovations | 15-RMD-UCHC-01 | Paul Robson |

The funders had no role in study design, data collection and interpretation, or the decision to submit the work for publication.

### Author contributions

Laura E Mickelsen, Formal analysis, Validation, Investigation, Visualization, Methodology, Writing - review and editing; William F Flynn, Data curation, Software, Formal analysis, Investigation, Visualization, Methodology, Writing - review and editing; Kristen Springer, Lydia Wilson, Eric J Beltrami, Investigation; Mohan Bolisetty, Software, Investigation, Methodology; Paul Robson, Resources, Data curation, Software, Supervision, Funding acquisition, Writing - review and editing; Alexander C Jackson, Conceptualization, Resources, Supervision, Funding acquisition, Investigation, Visualization, Writing - original draft, Project administration, Writing - review and editing

### Author ORCIDs

William F Flynn https://orcid.org/0000-0001-6533-0340
Paul Robson https://orcid.org/0000-0002-0191-3958
Alexander C Jackson https://orcid.org/0000-0001-7489-3946

### Ethics

Animal experimentation: All experiments were performed in accordance with the ethical guidelines described in the National Institutes of Health Guide for the Care and Use of Laboratory Animals and were approved by the Institutional Animal Care and Use Committee of the University of Connecticut and of the Jackson Laboratory for Genomic Medicine. All surgery was performed under ketamine/xylazine anesthesia and every effort was made to minimize suffering.

### Decision letter and Author response

Decision letter https://doi.org/10.7554/eLife.58901.sa1
Author response https://doi.org/10.7554/eLife.58901.sa2

## Additional files

### Supplementary files

• Transparent reporting form

### Data availability

FASTQ files and unfiltered count matrices for the 10X libraries: https://www.ncbi.nlm.nih.gov/geo/query/acc.cgi?acc=GSE146692. Code to generate figures and produce the analysis: https://github.com/TheJacksonLaboratory/ventral-posterior-hypothalamus-scrnaseq (copy archived at https://github.com/elifesciences-publications/ventroposterior-hypothalamus-scrna-seq). Analyzed, aggregated scRNA-seq object: https://singlecell.jax.org/hypothalamus.

The following dataset was generated:

| Author(s) | Year | Dataset title | Dataset URL | Database and Identifier |
|---|---|---|---|---|
| Flynn WF, Mickelsen LE, Robson P, Jackson AC, Springer K, Beltrami EJ, Bolisetty M, Wilson L | 2020 | Single cell RNA sequencing to classify molecularly distinct neuronal and non-neuronal cell types in the mouse ventral posterior hypothalamus | https://www.ncbi.nlm.nih.gov/geo/query/acc.cgi?acc=GSE146692 | NCBI Gene Expression Omnibus, GSE146692 |

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
