## [Decision Letter]

**Acceptance summary:**

These studies provide a single-cell transcriptomic analysis of the ventral posterior hypothalamus and were validated using fluorescent in situ hybridization. These data sets will be a valuable resource to the field of hypothalamic biology.

**Decision letter after peer review:**

Thank you for submitting your article "Cellular taxonomy and spatial organization of the murine ventral posterior hypothalamus" for consideration by *eLife*. Your article has been reviewed by three peer reviewers, one of whom is a member of our Board of Reviewing Editors, and the evaluation has been overseen by Gary Westbrook as the Senior Editor. The reviewers have opted to remain anonymous. The reviewers have discussed the reviews with one another and the Reviewing Editor has drafted this decision to help you prepare a revised submission.

We would like to draw your attention to changes in our revision policy that we have made in response to COVID-19 (https://elifesciences.org/articles/57162). Specifically, when editors judge that a submitted work as a whole belongs in *eLife* but that some conclusions require a modest amount of additional new data, as they do with your paper, we are asking that the manuscript be revised to either limit claims to those supported by data in hand, or to explicitly state that the relevant conclusions require additional supporting data. Our expectation is that the authors will eventually carry out the additional experiments and report on how they affect the relevant conclusions either in a preprint on bioRxiv or medRxiv, or if appropriate, as a Research Advance in *eLife*, either of which would be linked to the original paper.

Summary:

Mickelsen and colleagues describe results from a series of studies using single cell sequencing to access the molecular identity of both neuronal and non-neuronal cell types of the mammillary body. The authors report 18 transcriptionally distinct non-neuronal cell types, and 20 transcriptionally distinct neuronal subtypes corresponding to cells residing in the premammillary, supramammillary, tuberomammillary, and arcuate nuclei as well as the lateral hypothalamic area and mammillary bodies. Furthermore, they performed projection mapping studies using anatomically-specific genetic markers for subdivisions of the mammillary bodies to demonstrate distinct projection patterns to the anterior thalamic nuclei. The study is well designed incorporating two independent, batch-corrected replicates and independent samples from both males and female mice in the analysis.

Essential revisions:

1) The paper leaves a considerable amount of work for a potential user of this information to understand what level of selectivity and specificity would be attained with combinations of the marker genes described in the paper. Based on the scRNA-Seq data, how well can different transcriptional clusters be separated at the level of individual cells in the dataset? How many genes are needed to achieve optimal separation?

2) A general criticism of the manuscript is that the authors have clearly captured neurons from multiple subnuclei of the ventral posterior hypothalamus (PM, SUM, TMN, ARC, LHA, and MB), but have not subclustered the cells from these different subnuclei. Given that marker genes are identified by a "one-versus-rest" methodology, the markers found for each of these subnuclei will broadly mark neurons within these regions, but it will be difficult to identify unique cell types within a subnucleus. Thus, a more rigorous approach of isolating and reclustering some of their cell groups corresponding to a specific subnucleus should be undertaken (similar to what was already done for cluster 8, SUM neurons). At a minimum, reclustering PM and MB neurons on their own seems reasonable; however, reclustering of ARC (Campbell et al., 2017) and LHA (Mickelsen et al., 2019; Rossi et al., 2019) neurons is unnecessary as previous publications have gone into detail for these structures.

3) Although the authors discuss at length the possibility the *Slc17a6+* / *Slc32a1+* neurons of VPHGLUT cluster 8 might represent SUM neurons know to project to dentate gyrus and corelease glutamate and GABA. If this cannot be addressed experimentally, the authors need to temper the conclusions that can be drawn here.

4) There is a remaining question about the dynamic range of the data used for classification. In neurons, Figure 2E (note: this is mislabeled as Figure 3E in the text) indicates that there are only a small multiple of UMIs over the number of detected genes. A histogram that summarizes the number of UMIs/gene in this dataset would be helpful. The authors should discuss the significance of the fairly low number of UMIs for most genes. In addition, do the authors know if this presumed low coverages extends to the marker-genes.

5) The authors could perform FISH analysis with combinations of marker genes to attempt to assess if they have a substantial false negative rate in detected genes. This could be established by measuring marker-gene co-expression ratios from FISH and comparing this values to the co-expression ratios predicted with their scRNA-seq datasets. Short of this, the technical issues need to be discussed in more detail.

---

## [Author Response]

Essential revisions:1) The paper leaves a considerable amount of work for a potential user of this information to understand what level of selectivity and specificity would be attained with combinations of the marker genes described in the paper. Based on the scRNA-Seq data, how well can different transcriptional clusters be separated at the level of individual cells in the dataset? How many genes are needed to achieve optimal separation?

Thank you for raising these important points. First, in our interpretation and visualization of the scRNA-seq data, we have been careful to present the marker genes describing each discussed population in the form of violin plots, which give a visual representation of each marker within each transcriptionally distinct population. Second, for as many marker genes as possible, we show their expression per cell in the form of UMAP plots, which show how robust and selective the expression of each maker is within the entire population, at single cell resolution. For selected clusters, we have chosen to highlight markers that are (i) specifically and/or robustly expressed in the cluster, (ii) present in the Allen Brain Atlas ISH database to provide spatial information, and (iii) biologically significant in the context of the existing literature. Third, to address the reviewers’ concerns about the sensitivity and specificity of the presented marker genes, we have uploaded marker gene lists for each level of clustering to this manuscript’s associated GitHub repository. Therein, for each gene list, we specify the fraction of cells in the marked cluster which expresses the gene (sensitivity) and the fraction of cells not in the marked cluster (specificity).

Nonetheless, there remains the unanswered question of how well these marker genes can separate *individual cells* from one another. To answer this, we tackled the related question – can we build a model that can predict the transcriptional cluster of a cell, and if so, how many genes are needed to perform accurate classification? Briefly, we randomly split our neuronal expression matrix and associated cluster labels into two sets: training (70%) and testing (30%), where cells from each cluster are proportionately represented in each set. Using the cells in the training set, we computed the top-k (k in (2, 3, 5, 10, 15, 20)) genes for each cluster based on the one-vs-rest log-fold-change of each gene in each cluster, yielding a set of genes of size <= 20*k (as there are 20 neuronal clusters). Then, for each set of genes corresponding to a value of k, the training set expression matrix was subset to only include these genes which we used to construct a random forest classification model; this yields one model for each value of k. Each model takes as input a subset of a cell’s gene expression signature and outputs a cluster label. We used each model to predict the cluster labels of the cells in the testing set, and then compared the predicted labels to the original clusters presented in the main text. This process is illustrated in a new supplementary Figure 1—figure supplement 3.

In this figure, we show that with as few as 80 total genes (5 genes per cluster, with overlap), we can predict the cluster identity of a neuron with over 90% accuracy (Figure 1—figure supplement 3B-D). Moreover, Figure 1—figure supplement 3E shows most of the misclassifications of our model are within related groups (e.g. GLUT4/GLUT5, or GLUT9/GLUT10 and GLUT7). Some populations, such as HA17 (TMN) and GLUT20 (LM) can be identified with as few as 2 genes per cluster (Figure 1—figure supplement 3G). This is consistent with the violin plots of Figure 2D in which many of the neuronal populations presented can be uniquely identified by the expression of one or two marker genes.

2) A general criticism of the manuscript is that the authors have clearly captured neurons from multiple subnuclei of the ventral posterior hypothalamus (PM, SUM, TMN, ARC, LHA, and MB), but have not subclustered the cells from these different subnuclei. Given that marker genes are identified by a "one-versus-rest" methodology, the markers found for each of these subnuclei will broadly mark neurons within these regions, but it will be difficult to identify unique cell types within a subnucleus. Thus, a more rigorous approach of isolating and reclustering some of their cell groups corresponding to a specific subnucleus should be undertaken (similar to what was already done for cluster 8, SUM neurons). At a minimum, reclustering PM and MB neurons on their own seems reasonable; however, reclustering of ARC (Campbell et al., 2017) and LHA (Mickelsen et al., 2019; Rossi et al., 2019) neurons is unnecessary as previous publications have gone into detail for these structures.

Thank you for this valuable feedback. Although we identified suites of markers that appear to broadly define neurons within a particular subregion of the VPH (ex. PM, SUM, TMN, Arc, LHA, and MB), the many of the clusters we identified in fact correspond to very specific cell types within heterogeneous anatomical subregions. For example, among neurons in the first iteration of clustering, we identified 20 distinct clusters, of which one corresponds to Arc KNDy neurons (*Kiss1*+, *Tac2*+, *Pdyn*+) (Figure 4—figure supplement 2). The markers that define this Arc population are not global makers for the Arc but for only one highly specific cell type within the Arc. This finding is consistent with previous single cell analysis of the Arc in isolation, in which KNDy neurons emerged as only one of twenty-four distinct Arc neuronal clusters (Campbell et al., 2017). Another example is the PM in which we found that at least two of the twenty neuronal clusters corresponded to anatomically defined PM subpopulations including the PMv (cluster 7) and PMd (cluster 6).

The most salient example is the MB, which is the focus of this work (Figures 6-9, Figure 6—figure supplement 1, Figure 7—figure supplement 1 and Figure 8—figure supplement 1). Previous scRNA-seq analysis of the whole mouse hypothalamus identified a single cluster (defined by co-expression of *Foxb1, Cck, Adcyap1* and *Cartpt*) ascribed to the MB (Chen et al., 2017), unsurprising given the large area of tissue collected but low sampling per anatomical subregion. In our work, by contrast, six of the twenty neuronal clusters in the first iteration of clustering corresponded to putative MB subpopulations; one LM population (cluster 20) and five medial mammillary populations (clusters 1-5). Although we identified global markers for the MB as a whole, as well as markers that were common to all five medial mammillary populations, our analysis did reveal highly distinct cell populations, defined by unique sets of markers, which also exhibited remarkable anatomical parcellation across anatomical subdomains of the MB.

Although we were quite surprised at the high degree of anatomical resolution provided by just two levels of clustering (global and neurons-only), we fully agree that further subclustering of some of these major clusters would indeed reveal biologically important neuronal cell types beyond the ones that we identified in our first submission. To that end, we have since performed a significant amount of additional analysis illustrated in seven new supplementary figures, which show subclustering analysis of the PMv (Figure 3—figure supplement 1), SUM (Figure 4—figure supplement 1), Arc (Figure 4—figure supplement 2), LHA/Tub (Figure 4—figure supplement 3), TMN (Figure 5—figure supplement 1), LM (Figure 6—figure supplement 1) and medial mammillary (Figure 8—figure supplement 1) regions. In most cases, the subclustering analyses preserve much of the global structure shown in Figure 2B but provide higher resolution annotation of each subpopulation. We have also added another new figure, Figure 10, which summarizes the relationships between transcriptionally-defined neuronal clusters in our scRNA-seq analysis, with hierarchical clustering, and anatomical subregions. We have also amended the text to include discussions of these in-depth subregion analyses. We believe that this additional, detailed analysis significantly strengthens the biological insights that may be derived from our data.

3) Although the authors discuss at length the possibility the Slc17a6+ / Slc32a1+ neurons of VPHGLUT cluster 8 might represent SUM neurons know to project to dentate gyrus and corelease glutamate and GABA. If this cannot be addressed experimentally, the authors need to temper the conclusions that can be drawn here.

Thank you for pointing this out. Our goal in linking the *Slc17a6+* / *Slc32a1+* population we found within a neuronal cluster corresponding to the SUM, as with many other clusters we describe, was to connect the transcriptionally distinct VPH clusters that our analysis revealed to the known biology of VPH cells and circuits. However, given that we cannot definitely identify this cluster as the dentate-projecting dual phenotype SUM population without additional functional experiments, we have adjusted the description of this section in the Results (subsection “A neuronal population in the supramammillary (SUM) nuclei”) as suggested.

4) There is a remaining question about the dynamic range of the data used for classification. In neurons, Figure 2E (note: this is mislabeled as Figure 3E in the text) indicates that there are only a small multiple of UMIs over the number of detected genes. A histogram that summarizes the number of UMIs/gene in this dataset would be helpful. The authors should discuss the significance of the fairly low number of UMIs for most genes. In addition, do the authors know if this presumed low coverages extends to the marker-genes.

Thank you, this is another important point concerning scRNA-seq data used for classification (also we corrected the typo). The small multiple of UMIs over genes in our study is consistent with numerous other single cell studies utilizing this technology (please see Author response table 1 below). Nonetheless, to address several of the concerns raised by the reviewers, we have provided new panels to Figure 1—figure supplement 2E-G. Figure 1—figure supplement 2E shows histograms of the number of unique transcripts per gene for various sets of genes as suggested by the reviewers. Practically all genes used as marker genes for various subpopulations are present with at least 1,000 UMIs, and typically with 10,000s of UMIs in the dataset. Marker genes are only present in relatively small portions of the overall dataset (as they are only present in select subsets of cells), and yet marker genes have more total transcripts expressed than ubiquitously expressed genes, such as cell-cycle genes (Figure 1—figure supplement 2F). We selected a few discriminatory marker genes that define a selection of neuronal clusters to show the distribution of UMIs/cell for those genes in the marked populations (blue) and other populations (gray) (Figure 1—figure supplement 2G). The mean expression of each gene in the marked populations (shown as a dashed black line) is at least 10 counts per cell, and often times much higher. Therefore, the UMIs per cell for marker genes in the cell type they mark is much higher than the overall UMIs/gene ratio found in the entire data set.

It is often suggested that drop-out (or the apparent sparsity) of droplet-based scRNA-seq data limits its ability to detect specific cell types or markers. The reported sensitivity of the 10X Genomics 3’ scRNA-seq assay is ~15% and ~30% for v2 and v3 chemistry, respectively (for reference, please refer to https://kb.10xgenomics.com/hc/en-us/articles/360001539051-What-fraction-of-mRNA-transcripts-are-captured-per-cell-). Given the 15% minimum specificity of the assays presented here, we would expect the typical marker gene to be robust to dropouts. In other words, we expect that if a population of neurons is present in this dataset, the sparsity of the data would not limit our ability to detect the presence of marker genes in some, if not all, cells of that population.

We can estimate how much of each transcriptomic library we’re able to capture using the 10X platform. Unique molecular identifiers (UMIs) allow for the detection and removal of PCR duplicates, and many duplicate reads containing identical (cell-barcode, UMI) pairs are discarded through this deduplication process. Sequencing saturation is a measure that describes the level of read duplication in a sequencing library and gives an estimate of how much of the library’s complexity is captured. Saturation is described by the equation S=1–UR where U is the total number of UMIs per library and R is the total number of reads per library. A value of 50% signifies that, at the current sequencing depth, each unique cell-transcript pair is associated with 2 reads on average. Similarly, 90% sequencing saturation means each cell-transcript pair is associated with 10 reads. As sequencing saturation increases, the likelihood of observing new information decreases and, viewed this way, sequencing saturation is a proxy for how much of the information in an entire library has been captured. The median per-cell sequencing saturation in our dataset is 66% (IQR: 59%-67%), meaning each transcript is accompanied by 3 reads on average, which is typical for high-complexity datasets like ours.

Please note that once cells are defined in clusters, the ensemble average of all expression across individual cells within that cluster can be used to represent the transcriptome of that cell type. This pooling would alleviate any concerns about missing lowly expressed genes as, if there are sufficient numbers of cells represented in the cluster, all expressed genes should be detected, though not necessarily in all cells (i.e. false negatives).

Lastly, our data shows similar or higher median genes/neuron and median UMIs/neurons as compared with previously published hypothalamic and other murine brain scRNA-seq datasets using similar technology. Please see Author response table 1 below:

**Author response table 1. resptable1:** 

Median Genes/neuron	Median UMIs/neuron	Brain region	Study
4,335	12,035	Hypothalamus (VPH)	Mickelsen et al., (this study)
3,034	6,897	Hypothalamus (preoptic)	Moffitt et al., Science, 2018 (doi: 10.1126/science.aau5324)
3,442	8,791	Hypothalamus (LHA)	Mickelsen et al., Nature Neuroscience, 2019 (doi: 10.1038/s41593-019-0349-8)
<4,000	<8,000	Hypothalamus (LHA)	Rossi et al. Science, 2019(doi: 10.1126/science.aax1184)
~3,000	~6,000	Hypothalamus (VMHvl)	Kim et al., Cell, 2019(doi: 10.1016/j.cell.2019.09.020)
<2,000	<3,000	Hypothalamus (SCN)	Wen et al. Nature Neuroscience, 2020(doi: 10.1038/s41593-020-0586-x)
1,097	1,688	Hypothalamus	Romanov et al. Nature, 2020(doi: 10.1038/s41586-020-2266-0)
2,679	6,570	E18 Cortex + Hippocampus	10X Genomics 10k Neurons Dataset (https://support.10xgenomics.com/single-cell-gene-expression/datasets/3.0.0/neuron_10k_v3)
1,927	4,226	E18 Cortex + Hippocampus	10X Genomics 1M Neurons Dataset (https://support.10xgenomics.com/single-cell-gene-expression/datasets/1.3.0/1M_neurons)

5) The authors could perform FISH analysis with combinations of marker genes to attempt to assess if they have a substantial false negative rate in detected genes. This could be established by measuring marker-gene co-expression ratios from FISH and comparing this values to the co-expression ratios predicted with their scRNA-seq datasets. Short of this, the technical issues need to be discussed in more detail.

Thank you for this comment. During this time, we are severely limited in our ability to carry out new FISH experiments owing to COVID19 restrictions but recognize the necessity of discussing the technical issues related to FISH in more detail. Both FISH and scRNA-seq, while powerful tools for analyzing cell types in complex tissues, have several technical limitations. scRNA-seq is the less sensitive method, estimated at ~15%, and therefore frequently having false negative cells for lowly expressed genes. Our selection of marker genes, as discussed above, targets genes that are generally more highly expressed, with a minimum of at least 10 counts per cell (but often much higher). At 15% sensitivity this would equate to a minimum of ~66 RNA molecules of the marker gene per intact cell, for the cell type that it is a marker for. The FISH technology we use here (RNAScope) requires two oligos to hybridize immediately adjacent to one another (zero bases in between) on the same RNA to subsequently build the probe structure from which the fluorescent signal is visualized (Wang et al., 2012). This two-oligo design greatly reduces signal-to-noise ratio. Twenty sets of such oligo pairs are designed across a single RNA molecule, and the amplified signal for each pair contains 100s of fluorescent molecules and thousands across the individual RNA molecule. We are confident that if the RNA molecule is present in the tissue section, RNAscope will detect it, and we expect the false negative rate to be minimal, and substantially lower than scRNAseq.

In this work, we are primarily using FISH to validate the co-expression findings from our scRNAseq dataset with the criteria for co-expression being a binary assessment of presence/absence of specific marker genes. While these criteria are not as quantitative as single molecule resolution, the presence of false negatives should be evident in our ratios of co-expression quantified in Figures 3-6. While RNAscope has the ability to detect single molecules, for many of our marker genes this is a challenge as the fluorescent signal in a positive cell can be overwhelming due to some many RNA molecules of the marker gene present. If false negatives are a significant confounder in our dataset, we would expect to see substantial discrepancies between the co-expression predicted by our single-cell transcriptomic data and the measured co-expression in our FISH data even on the level of presence/absence, which is not the case. Overall, we see a substantial concordance between our FISH and scRNA-seq data, which suggests that false negatives are unlikely to be a major confounder in our analysis.